# Planar cell polarity proteins mediate ketamine-induced restoration of glutamatergic synapses in prefrontal cortical neurons in a mouse model for chronic stress

Andiara E. Freitas [1], Bo Feng[1], Timothy Woo [1], Shae Galli[1], Clayton Baker [1], Yue Ban[1], Jonathan Truong[1], Anna Beyeler [2] & Yimin Zou [1] ✉

Single administration of low-dose ketamine has both acute and sustained anti-depressant effects. Sustained effect is associated with restoration of glutamatergic synapses in medial prefrontal cortic (mFPC) neurons. Ketamine induced profound changes in a number of molecular pathways in a mouse model for chronic stress. Cell-cell communication analyses predicted that planar-cell-polarity (PCP) signaling was decreased after chronic administration of corticosterone but increased following ketamine administration in most of the excitatory neurons. Similar decrease of PCP signaling in excitatory neurons was predicted in dorsolateral prefrontal cortical (dl-PFC) neurons of patients with major depressive disorder (MDD). We showed that the basolateral amygdala (BLA)-projecting infralimbic prefrontal cortex (IL PFC) neurons regulate immobility time in the tail suspension test and food consumption. Conditionally knocking out Celsr2 and Celsr3 or Prickle2 in the BLA-projecting IL PFC neurons abolished ketamine-induced synapse restoration and behavioral remission. Therefore, PCP proteins in IL PFC-BLA neurons mediate synapse restoration induced by of low-dose ketamine.

Synapse atrophy in the prefrontal cortex in patients of major depressive disorder (MDD) is well documented[1]. A non-competitive NMDA antagonist, ketamine, has been shown to have both fast-acting and sustained antidepressant effects[2]. The fast-acting effect occurs several hours after treatment, whereas the sustained effect, likely triggered by fast-acting effects, can last for more than 7 days[3,4]. The mechanisms of action for both the fasting-acting and sustained effects are not well understood, especially the long-term sustained antidepressant effects. Using in vivo two photon imaging, it was found that some of the glutamatergic synapses in the anterior cingulate cortex were lost in animal models for chronic stress and depression but restored by ketamine along with new synapses starting from a few hours after treatment[5]. The profound effect of

ketamine on synapse restoration and synaptogenesis suggests that ketamine may trigger wide-range changes of cell–cell signaling in multiple neuronal types and non-neuronal cell types. To begin to explore the mechanisms, we performed single-cell sequencing in medial prefrontal cortex (mPFC) and basolateral amygdala (BLA) of animals treated with corticosterone, ketamine and ketamine following corticosterone. Using CellChat, we found that many signaling pathways were predicted to be altered and ketamine reversed many of the changes induced by chronic administration of corticosterone[6]. Planar cell polarity signaling, which plays pivotal roles in glutamatergic synapse formation and maintenance, was predicted to be decreased in excitatory neurons in mPFC in animals treated with corticosterone as well as in patients with major depressive disorder

[1]Department of Neurobiology, School of Biological Sciences, University of California, San Diego, La Jolla, CA 92093, USA. [2]Neurocentre Magendie, University of Bordeaux, 146, Rue Leo Saignat, 33000 Bordeaux, France. ✉e-mail: yzou@ucsd.edu

(MDD)[7–10]. And low-dose ketamine increased planar cell polarity signaling in most of the excitatory neurons in animals treated with corticosterone. Optogenetic activation of infralimbic medial prefrontal cortex (IL PFC) neurons was shown to mimic the ketamine-induced long-term antidepressant effect[11]. We found that the BLA-projecting IL PFC neurons mediate anti-depressant-like function. Using CRISPR–Cas9, we conditionally knocked out the PCP genes which are required for glutamatergic synapse formation and found that ketamine-induced synapse restoration in these neurons and the behavioral remission were abolished.

## Results

### Single-cell transcriptomics, cell–cell communication and gene ontology analyses in the mPFC and BLA of mice treated with corticosterone and ketamine

To explore the mechanisms underlying the long-term synapse changes induced by chronic stress and ketamine treatment, we performed single-cell transcriptomics analyses in the mPFC and BLA of animals treated with corticosterone or low-dose ketamine and those first treated with corticosterone then with ketamine. C57BL/6 J mice were exposed to corticosterone (CORT, 35 μg/ml; equivalent to 5 mg/kg/day) or control (2-hydroxypropyl cyclodextrin, BCD) for 6 weeks in their drinking water and then injected with ketamine (Ket, 10 mg/kg, intraperitoneal, i.p.) or saline. 24 h after Ket or saline treatments, brains were harvested, and the mPFC and BLA tissues were dissected, and single cells were isolated and processed for sequencing. A diagram of all experimental schedules is given in Fig. 1a. A total of 42,475 single cells from 10 independent experimental samples were sequenced. An average of 14,465.1 genes/sample was captured. As described in the methods section, all samples were submitted to quality control thresholds, of which cells with a high percentage of mitochondrial mRNA (>12.5%, an indicative of unhealthy cells) were filtered out. This resulted in a total of 13,381 high quality cells. The cells were then subjected to the standard Seurat processing pipeline as described in methods, which resulted in 17 clusters (Fig. 1b). The cluster specific cell markers were identified by a comparative analysis looking at the top changing genes by their average expression fold change and cluster specificity: percent expression in a given cluster compared to all other clusters. Cell type-specific gene marker expression in the cell clusters is given in Supplemental Fig. 1 as a heatmap.

To understand the changes ketamine treatment induces, we performed cell–cell communication analyses based on receptor ligand pair gene expression with the CellChat package and an enriched database for each treatment type and region[6]. For each group an inferred cell-to-cell communication network was formed. Next, we performed a comparison of the inferred cell-to-cell communication networks for the treatment groups for mPFC and BLA. We found that many of the same cell–cell communication pathways that were predicted downregulated in corticosterone-treated mice were upregulated after ketamine treatment. Most of the pathways which showed the greatest predicted reversal in the corticosterone-treated animals by ketamine treatment are cell–cell interacting molecules, including cell adhesion pathways that may impact neuronal morphology and structure (Fig. 1c). CDH and VTN were the pathways that were predicted to be reduced most by corticosterone but increased the most following ketamine administration of the corticosterone-treated animals in the mPFC. SEMA6, EGF, NRG, L1CAM, NEGR, NRXN, VCAM, CADM, CNTN, and NCAM were the names of the pathways that were predicted to be reduced most by corticosterone but increased the most following ketamine administration of the corticosterone-treated animals in the BLA. Stress and depressive conditions are associated with neuroinflammation. Inflammatory signatures induced by corticosterone treatment were also greatly decreased by ketamine treatment. The pathways that underwent significant predicted reduction included chemokines C-X-C motif ligand (CXCL), macrophage

migration inhibitory factor (MIF) and colony-stimulating factors CSF) (Fig. 1d). Up regulation of the FGF pathway in different cell types, especially in astrocytes, is proinflammatory. The predicted increase in FGF signaling induced by corticosterone was reversed by ketamine (Supplemental Fig. 2a). Consistent with CellChat analyses, gene ontology (GO) analyses predicted similar opposite changes of many pathways, of which top 10 were listed (Supplemental Fig. 2b, c). For instance, the expression of two key pro-inflammatory genes, IL-1ß and NFkB, were found increased in the MAP kinase pathway (Supplemental Fig. 2d) in corticosterone-administered animals (ranked top 5 in Supplemental Fig. 2b) and the NFkB signaling was found reduced in the MAP kinase pathway (Supplemental Fig. 2e) after ketamine treatment of corticosterone-administered animals (ranked top10 in Supplemental Fig. 2c). To validate this at the protein level, we then performed immunostaining and the results confirmed that corticosterone did induce neuroinflammation, with an increase of IL-1ß and NFkB in microglia, and ketamine reversed these effects (Fig. 1e–h).

### Changes of planar cell polarity (PCP) signaling in the mPFC and BLA after ketamine treatment of chronic corticosterone-exposed mice

Planar cell polarity (PCP) proteins are key regulators of glutamatergic synapse formation and maintenance[8,12]. PCP, the global cell and tissue polarity along the tissue plane, is a fundamental organizational property of many tissues[13,14]. The 6 conserved core PCP signaling components form asymmetric protein complexes at the Cadherin-mediated adherens junctions that connect and polarize neighboring cells[13,14]. Components of the PCP pathway are localized in developing excitatory synapses, in similar fashions as during PCP signaling at cell–cell junctions, and interact with multiple key presynaptic and postsynaptic proteins. Celsr3 is one of three mouse homologs of Flamingo, a core PCP component which brings adjacent cells together. Using electron microscopy, immunocytochemistry and electrophysiology, it was shown that conditionally knocking out Celsr3 in hippocampal pyramidal neurons at postnatal day 7 led to a reduction of approximately 50% of glutamatergic synapses in vivo when examined on postnatal day 14. These Celsr3 cKO animals showed behavioral deficits in spatial learning and fear memory in adulthood[7,8]. Conditionally knocking out Vangl2, another PCP component, which antagonizes the Frizzled/Dishevelled complex, led to approximately 50% increase of the glutamatergic synapse numbers in development[7,8]. More recent studies show that conditionally knocking out Celsr3 and Celsr2 in the adult hippocampus and mPFC led to a loss of 50% of glutamatergic synapses two months later, and conditionally knocking out Vangl2 in adulthood lead to a 27% increase of synapse numbers[10]. Another core PCP component, Prickle, is required for the formation and maintenance of at least 70–80% of the PSD-95-positive synapses in hippocampus and mPFC, and promotes synapse formation by antagonizing Vangl2[9].

In light of the important role of PCP pathway in glutamatergic synapse formation and maintenance, we analyzed the inferred changes of the PCP pathway using CellChat. The original CellChat database did not include the PCP pathway although it has the non-canonical Wnt pathway, ncWNT pathway. We added the key PCP components, based on our previous findings, into the interaction_input_CellChatDB file as Secreted Signaling or Cell–Cell contact and updated the DB slot within the CellChat object (Supplemental Fig. 3). Celsrs are adhesion G-protein coupled receptors (GPCRs) with cadherin repeats and are homophilic interacting proteins. In mPFC, Celsr2–Celsr2 interaction was predicted to be reduced among excitatory neurons but increased between Type 1 excitatory neurons and inhibitory neurons following corticosterone treatment. After ketamine treatment, the corticosterone-treated animals showed predicted increased Celsr3–Celsr3 interactions in Type 1 excitatory neurons but decreased Celsr2–Celsr2 interactions in Type 1 excitatory neurons and between

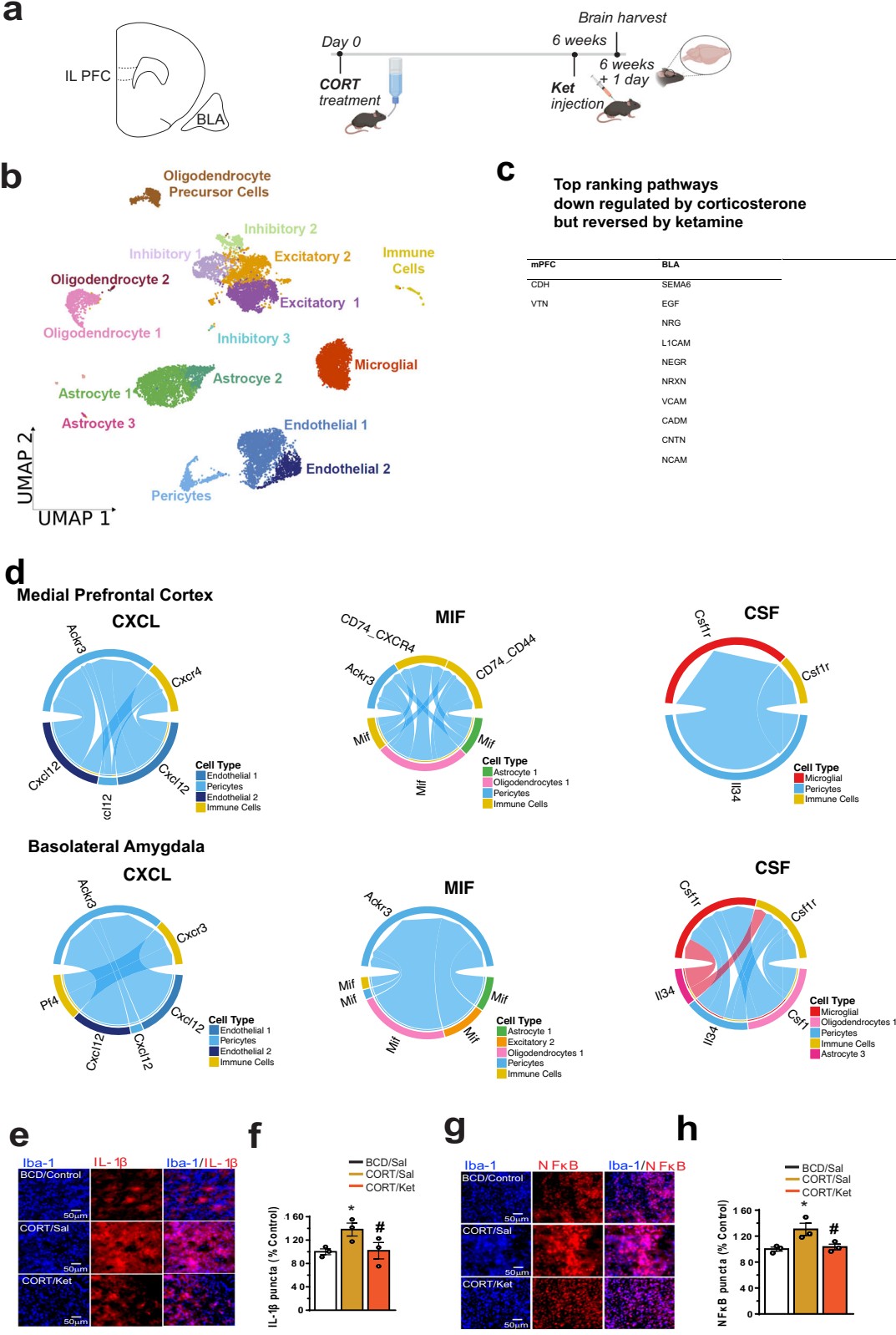

**Fig. 1 | Single-cell transcriptomic and cell–cell communication analyses.**
**a** Schematics of experimental design. The illustrations were created with BioRender. **b** UMAP plot of 17 Identified Cell Types. **c** Table of the top rankding pathways identified with CellChat that are inversely correlated between corticosterone-treated animals followed by ketamine treatment vs cortiosterone-treated animals without ketamine treatment. **d** CellChat diagrams made with a modified Chord-Diagram where blue indicates reduced expression level in corticosterone ketamine treated mice and red indicates increased expression level in corticosterone ketamine treated mice. **e, f** Immunostaining of IL-1ß in microglia [$F_{(2, 6)}$ = 5.474, $P$ = 0.044]. **g, h** Immunostaining of NFkB in microglia [$F_{(2, 6)}$ = 6.876, $P$ = 0.028]. Each column represents the mean + S.E.M. of 3 animals. Statistical analysis was performed by one-way ANOVA followed by Fisher's LSD. *$p < 0.05$ compared with the control group (BCD) treated with saline, and #$p < 0.05$ compared with the CORT treated with saline, Scale bar 50 μm.

## a

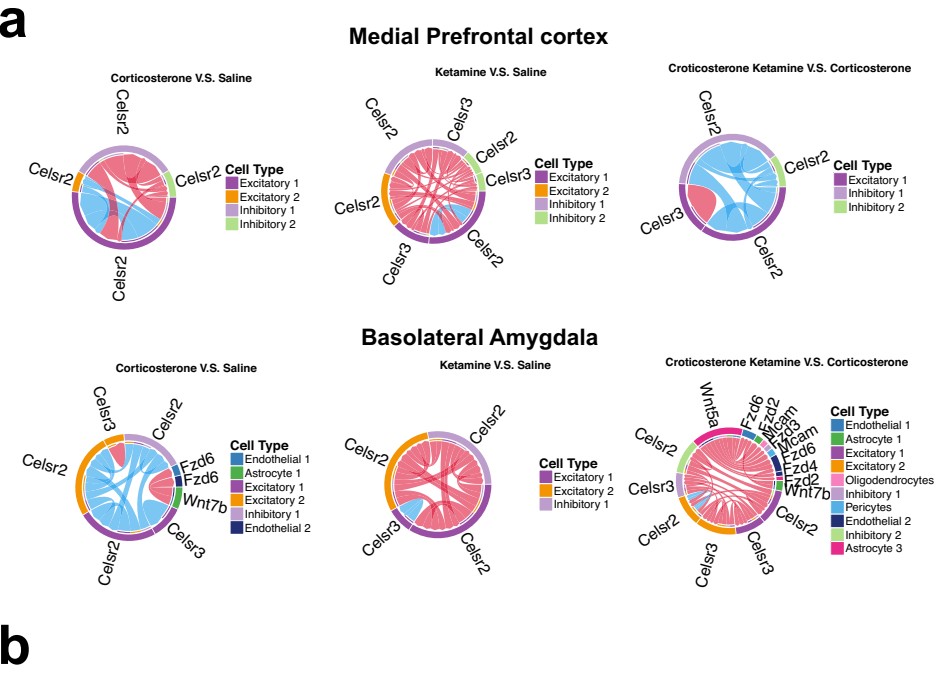

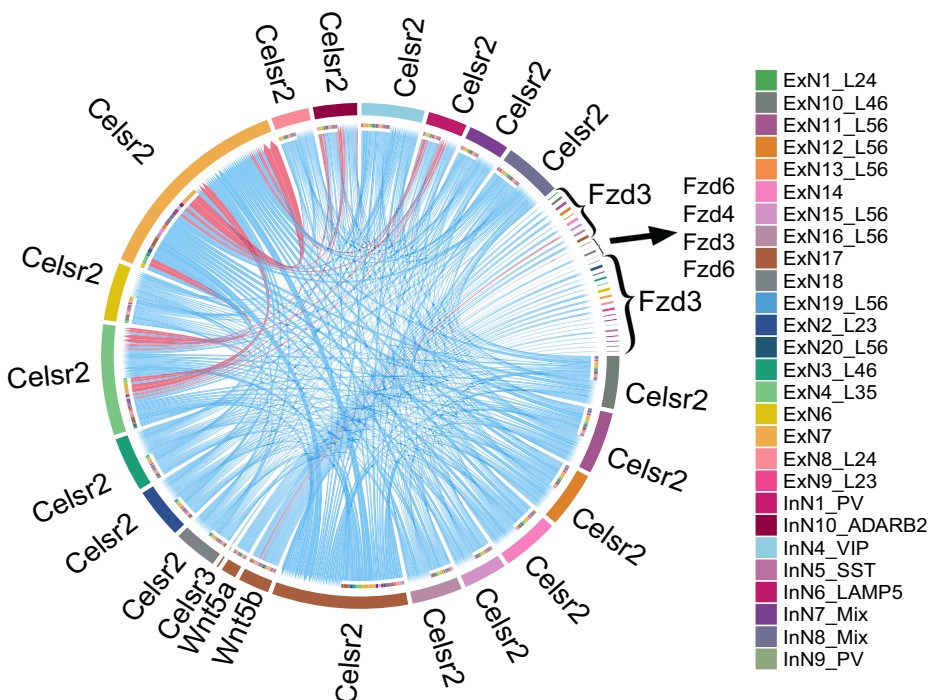

**Fig. 2 | Changes of PCP signaling inferred by cell-cell communication analyses. a** CellChat Diagrams of PCP signaling pathway made with a modified ChordDiagram where blue indicates reduced expression level and red indicates increased expression level in mice. **b** CellChat Diagrams of PCP signaling pathway made with a modified ChordDiagram where blue indicates reduced expression level in MDD patients and red indicates increased expression level in MDD patients.

Type1 excitatory neurons with inhibitory neurons in prefrontal cortex (Fig. 2a). In BLA, Celsr2–Celsr2 interactions and Celsr3–Celsr3 inter-action were predicted reduced in Type 1 excitatory neurons and inhi-bitory neurons, whereas Celsr3 is increased in Type 2 excitatory neurons after corticosterone treatment (Fig. 2a). Ketamine treatment of corticosterone-treated animals led to a predicted increase of Celsr3–Celsr3 interaction in all excitatory and inhibitory neurons in amygdala. Celsr2–Celsr2 interactions was predicted increased in nearly all neuronal types, except Type 2 excitatory neurons in BLA

following ketamine treatment (Fig. 2a). Therefore, PCP signaling shows potentially complex and cell type-specific patterns of changes fol-lowing corticosterone treatment and after subsequent ketamine treatment. Ketamine treatment without corticosterone treatment led to some opposite changes by corticosterone treatment and some similar changes with ketamine treatment after corticosterone admin-istration, again with highly complex cell-type-specific patterns of changes of PCP signaling among different neuronal cell types (Fig. 2a). To ask whether similar changes in the PCP pathway may also take place

in humans, we downloaded and analyzed the Major Depressive Disorder (MDD) Single Nucleus RNAseq Human Dataset from the UCSC cell browser, which included a total of over 160,000 nuclei from 71 dorsolateral-PFC samples[15]. An inferred cell–cell communication network analysis was performed for two groups: patients with and without MDD diagnosis. Differential network analysis was conducted between the two for the PCP pathway. Celsr2 and Celsr3 were predicted downregulated in most types of excitatory neurons in humans (Fig. 2b). Like in mouse mFPC, Celsr2 was predicted increased between excitatory neurons and inhibitory interneurons (PV, ADARB2). Interestingly, Celsr2 was predicted decreased between excitatory neurons and VIP interneurons. These analyses suggest that neuron cell type-specific or neural circuit-specific changes of PCP proteins are similar between the mouse model for chronic stress/depression and human MDD and may mediate specific synaptic changes in these neurons and circuits. The extent of changes of PCP gene expression is in general much greater in the amygdala than in the mPFC, consistent with the greater numbers of cell-cell signaling pathways that were predicted altered in the amygdala (Fig. 1b).

### IL PFC neurons projecting to BLA control immobility time in the tail suspension test (TST) and food consumption

Previous studies showed that optogenetic stimulation of infralimbic prefrontal cortex (IL PFC) neurons mimics the ketamine-induced long-term antidepressant-like effect[11]. IL PFC neurons project to several brain regions which are related to anxiety and negative valence, including BLA and anterior insular cortex (aIC)[16]. To determine which neuronal population mediates antidepressant-like effects, we expressed the Gi/o-coupled inhibitory DREADD in either the IL PFC neurons which project to basolateral ganglia (BLA) or anterior insular cortex (aIC) and submitted the animals to tail suspension test (TST) and measured food consumed in the food consumption test (FCT)[17]. In more details, C57BL/6 J mice were stereotactically injected bilaterally into the IL PFC with a Cre-dependent inhibitory DREADD hM4Di-mCherry virus or its mCherry control, and into the BLA or aIC with AAV.hSyn.Cre.WPRE.hGH (AAV Rretrograde). Three weeks after surgery, mice were treated with the chemogenetic ligand compound 21 (C21) at the dose of 2 mg/kg (i.p.) and, 40 min later, underwent a series of behavioral tests (a diagram of experimental schedule is given in Fig. 3a, b). As illustrated in Fig. 3c, d silencing the BLA-projecting IL PFC neurons caused an increase in immobility time in the TST. Conversely, the inhibition of the aIC-projecting IL PFC neurons did not cause any alteration in immobility time in the TST (Fig. 3h, i). Additionally, silencing either the BLA-projecting IL PFC (Fig. 3e, f) or aIC-projecting IL PFC (Fig. 3j, k) neurons did not alter mice locomotion in the open-field. Lack of interest is a symptom used to diagnose depression and it is characterized as the inability to feel pleasure during activities commonly appreciated by individuals[18]. Food consumption, partly driven by the reward of food intake, is often tested in the context of animal models for stress. We therefore performed food consumption test (FCT, see methods for details). The animals were trained in the FCT protocol and 24 h later their food consumption was measured. Chemogenetic inhibition of the IL PFC neurons projecting to the BLA caused a reduction in the food consumed (Fig. 3g), whereas the blockade of the IL-PFC neurons projecting to the aIC did not alter food consumption (Fig. 3l). Taken together, these data reveal that the BLA-projecting IL PFC neurons population, but not the aIC-projecting IL PFC neurons, control immobility time in the TST and food consumption in the FCT.

### Cell type-specific changes of PCP gene expression in the mPFC after chronic corticosterone administration and ketamine treatment

We then analyzed the cell-type-specific expression of core PCP genes in our single cell transcriptomics data (Supplemental Fig. 4a–c). Our single-cell sequencing results showed that chronic CORT administration led to a downregulation of Celsr2, Celsr3, and Prickle2 in excitatory neurons, which was then restored by a single injection of low-dose Ket. On the other hand, CORT downregulated Celsr3 and Prickle2, but upregulated Celsr2 in inhibitory neurons. Such alterations in interneurons were then reversed after Ket treatment. The violin plots of these PCP core genes are given in Supplemental Fig. 4b, c. Of note, the fast-acting effect of Ket has been associated with changes in excitatory and inhibitory transmission[19,20]. In our study, CORT treatment increased the expression of Celsr2 in inhibitory neurons and Ket treatment reversed that, suggesting that the alterations in inhibition may be mediated by Celsr2, whose expression is regulated differently in inhibitory and excitatory neurons. In order to validate our single-cell transcriptional results, we performed multiplex fluorescent RNAscope to measure the transcripts of the PCP genes in the mPFC. A diagram of experimental schedules is given in Fig. 4a. These results confirmed that *Celsr2*, *Celsr3*, and *Prickle2*, in excitatory neurons, were indeed downregulated by CORT but restored by Ket in CORT-treated animals (Fig. 4b–d, h–j). In agreement with our single cell sequencing data, the RNAscope results showed that *Celsr3* and *Prickle2*, but not *Celsr2*, in inhibitory neurons, were downregulated by CORT and such transcriptional changes were rescued by Ket (Fig. 4e–g). In fact, *Celsr2* was increased by CORT and then decreased by Ket in inhibitory neurons (Fig. 4e, f). Supplemental Fig. 5 illustrates the quality controls for RNAscope Multiplex Fluorescent Assay.

### Celsr2/3 and Prickle 2 are essential in the BLA-projecting IL PFC neurons for ketamine-mediated remission of depressive-like behaviors induced by CORT

In order to determine whether the PCP proteins may mediate ketamine-induced synapse restoration and behavioral remission, we first tested whether these PCP proteins are required for sustained antidepressant-like effect induced by ketamine. CRISPR–Cas9 mice were exposed to 35 μg/ml of CORT in the drinking water for three weeks, after the 3 weeks, mice were stereotactically injected bilaterally into the IL PFC with a combination of AAV-FLEX-EGFP and AAV-U6-sgRNA-h-Syn-mCherry, and into the BLA with AAV.hSyn.Cre.WPRE.hGH (AAV Rretrograde). Three weeks following surgery, mice were treated with Ket (10 mg/kg, i.p.) and 24 h later underwent a series of behavioral tests. The first test was the TST. Immediately after the TST, mouse locomotion was evaluated in the open field paradigm. The open field test is used to discard any possible false-positive antidepressant effect caused by a psychostimulant drug. 24 h after the TST, the feeding behavior of mice was evaluated. A diagram of all experimental schedules is given in Fig. 5a. As illustrated in Fig. 5b, c the chronic treatment of CRISPR–Cas9 mice with CORT caused a significant increase in the immobility time of mice in the TST, indicative of depressive-like behavior, which was reversed by Ket. Notably, knocking out *Celsr2/3* and *Prickle2* prevented Ket from revsersing the depressive-like behavior induced by CORT, showing that these PCP proteins are necessary for Ket's sustained antidepressant-like effects. Importantly, no locomotor alteration was observed in any experimental groups, indicating a specific antidepressant-like effect elicited by Ket (Fig. 5d, e). Our results show that by knocking out *Celsr2/3* or *Prickle* also prevented Ket's ability to increase reduced food consumption (Fig. 5f). Taken together, our behavioral data supports that the antidepressant-like effect of Ket requires *Celsr2/3* and *Prickle2* in the BLA-projecting IL PFC neurons. It is worth noting that knocking out *Celsr 2/3* or *Prickle2* in the BLA-projecting IL PFC neurons in control mice (*no CORT or Ket administration*) did not cause any behavioral alteration in the TST, open-field or FCT (Supplementary Fig. 7a–f) but lead to a decrease of spine/synapse density (Supplementary Fig. 7g–i). It is possible that when we knocked out *Celsr2/3* and *Prickle2*, we were reducing all synapses non-selectively, whether they were active (functional) or not, or

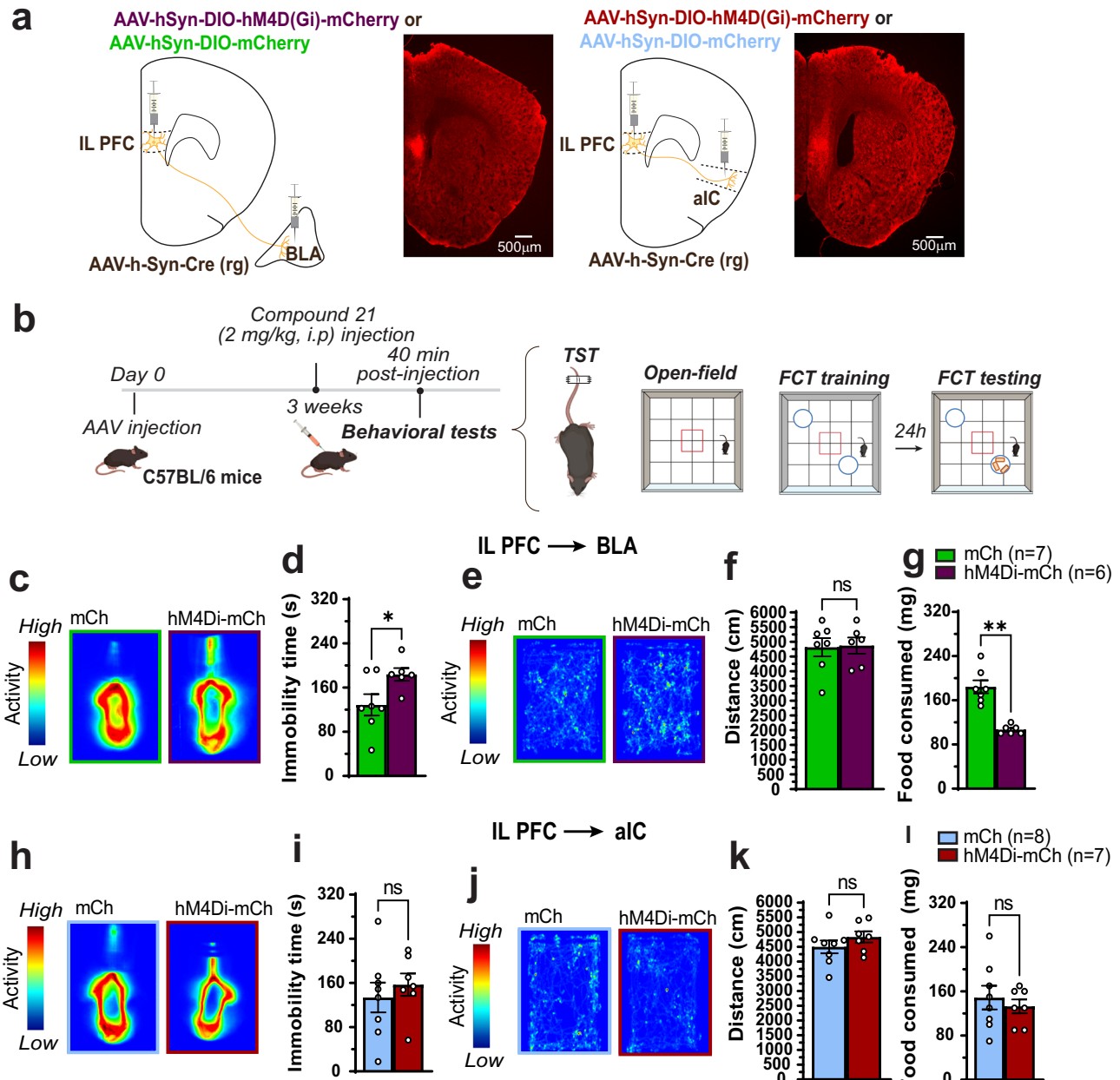

**Fig. 3 | IL PFC neurons projecting to the BLA control immobility time in the tail suspension test and food consumption in the food consumption test.**
**a** Representative micrographs validating viral injections (repeated at least twice). The illustrations were created with BioRender. **b** Schematics of experimental design. The illustrations were created with BioRender. **c** Representative heatmaps of mouse movement in the TST. **d** Quantification of immobility time in the TST in (**c**) [$T = 2.35$, $p = 0.038$], mCh control $n = 7$, hM4Di-mCh $n = 6$. **e** Representative heatmaps of the open field test. **f** Quantification of locomotion in the open field test in (**e**) [$T = 0.13$, $p = 0.89$], mCh control $n = 7$, hM4Di-mCh $n = 6$. **g** Quantification of food consumption in the food consumption test [$T = 5.92$, $p < 0.0001$], mCh control $n = 7$, hM4Di-mCh $n = 6$. **h** Representative heatmaps of mouse movement in the TST. **i** Quantification of immobility time in the TST in (**h**) [$T = 0.67$, $p = 0.51$], mCh control $n = 8$, hM4Di-mCh $n = 7$. **j** Representative heatmaps of the open field test. **k** Quantification of locomotion in the open field test in (**j**) [$T = 1.15$, $p = 0.27$], mCh control $n = 8$, hM4Di-mCh $n = 7$. **l** Quantification of food consumption in the food consumption test [$T = 0.61$, $p = 0.55$], mCh control $n = 8$, hM4Di-mCh $n = 7$. Each column represents the mean + S.E.M. of $n$ animals provided above. Statistical analysis was performed by Student's $t$-test two-tailed. *$p < 0.05$ and ****$p < 0.0001$ compared with the control group (mCh).

selectively those that were not active. Therefore, behaviors were not affected as enough of the functional synapses remained. Hence, our results suggest that these PCP proteins may be required for the formation of new synapses induced by low-dose ketamine, which are active and can support behavioral remission. The lack of sustained antidepressant like effect of ketamine in PCP gene knockout is not due to an overall decrease of synapses but due to the lack of the induction of new PCP gene expression by ketamine as the knockout itself does not cause depressive-like behavior.

**_Celsr2/3_ and _Prickle 2_ are required in the BLA-projecting IL-PFC neurons for ketamine-mediated restoration of dendritic spines and excitatory synapses in chronically stressed mice**
To directly test whether PCP proteins are required for the ketamine-induced synapse restoration, we examined the synapse numbers in the corticosterone-administered animals after ketamine treatment in _Celsr2/3_ or _PK2_ conditional knockout. Immediately after the behavioral tests mentioned in Fig. 5, all animals were perfused. Their brain tissues were then harvested and immunostained for excitatory synapses.

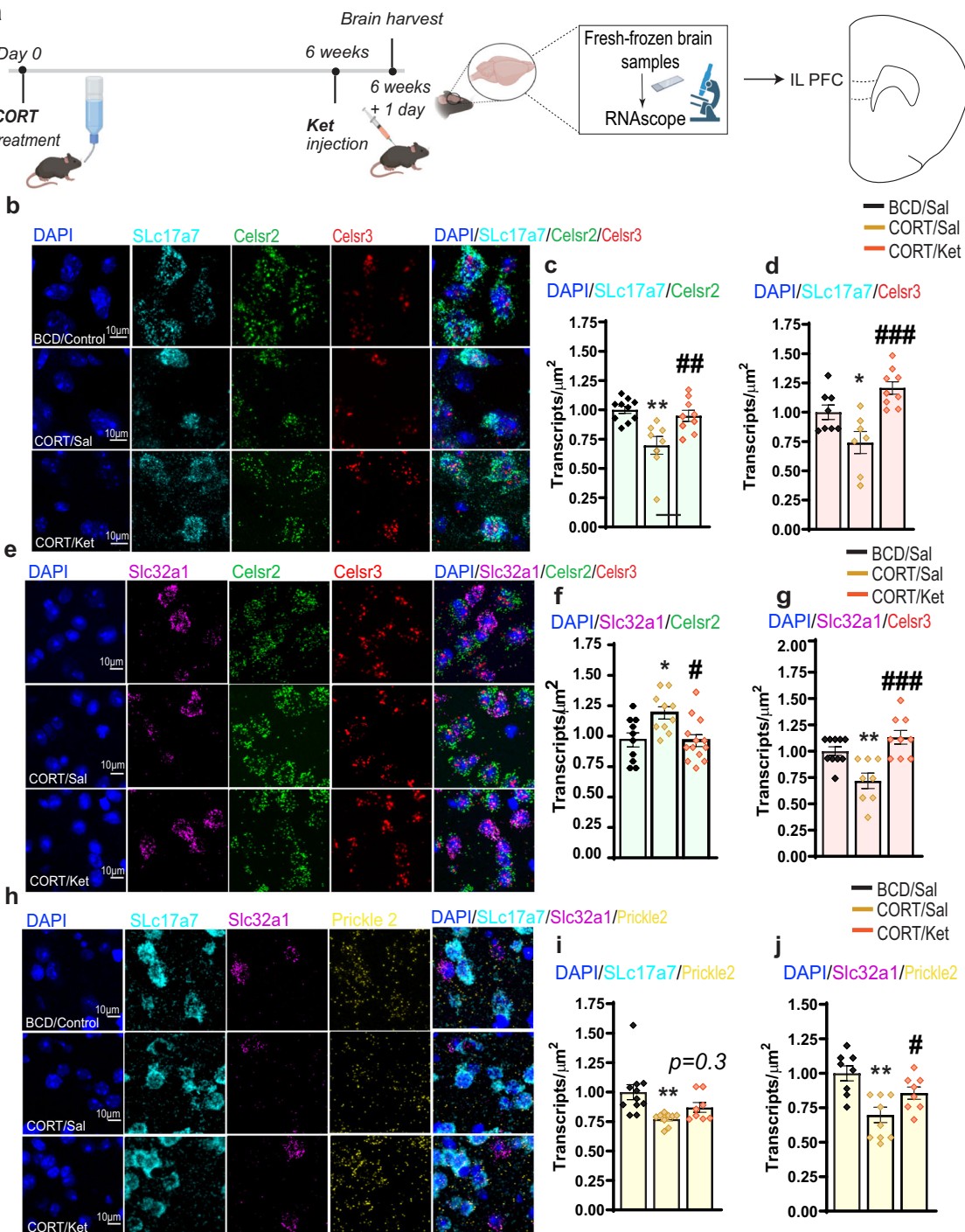

**Fig. 4 | Verification of PCP gene expression changes by RNAscope. a** Schematics illustrating the experimental design. The illustrations were created with BioRender. **b**, **e**, **h** Representative micrographs of RNAscope. **c** Quantification of *Celsr2* expression in excitatory neurons [$F_{(2, 24)} = 9.518$, $P = 0.0009$], BCD/Sal control $n = 10$, CORT/Sal $n = 8$, CORT/Ket $n = 9$. **d** Quantification of *Celsr3* expression in excitatory neurons [$F_{(2, 21)} = 11.36$, $P = 0.0005$], BCD/Sal control $n = 8$, CORT/Sal $n = 7$, CORT/Ket $n = 9$. **f** Quantification of *Celsr2* in inhibitory neurons [$F_{(2, 30)} = 5.829$, $P = 0.0073$], BCD/Sal control $n = 10$, CORT/Sal $n = 10$, CORT/Ket $n = 13$. **g** Quantification of *Celsr3* in inhibitory neurons [$F_{(2, 24)} = 11.88$, $P = 0.0003$], BCD/

Sal control $n = 10$, CORT/Sal $n = 8$, CORT/Ket $n = 9$. **i** Quantification of *Pricke2* in excitatory neurons [$F_{(2, 25)} = 5.804$, $P = 0.0085$], BCD/Sal control $n = 11$, CORT/Sal $n = 9$, CORT/Ket $n = 8$. **j** Quantification of *Pricke2* in inhibitory neurons [$F_{(2, 23)} = 8.506$, $P = 0.0017$], BCD/Sal control $n = 8$, CORT/Sal $n = 10$, CORT/Ket $n = 8$. Each column represents the mean + S.E.M. of $n$ fields provided above. Statistical analysis was performed by one-way ANOVA followed by Tukey's test. *$p < 0.05$ and **$p < 0.01$ compared with the control group (BCD) treated with saline, #$p < 0.05$, ##$p < 0.01$ and ###$p < 0.001$ compared with the CORT group treated with saline. Scale bar 10 μm; 50 μm; Magnification ×40.

---

Secondary and tertiary branches of the dendrites of the the BLA-projecting IL PFC cortex neurons were imaged with confocal microscope. A diagram of experimental schedules is given in Fig. 6a. As shown in Fig. 6b–d, chronic treatment of the control mice with CORT

caused a significant reduction of the number of dendritic spines (Fig. 6b, c) and excitatory synapses (Fig. 6b, d). A single administration of Ket restored the number of dendritic spines (Fig. 6b, c) and excitatory synapses (Fig. 6b, d) in control mice. Notably, knocking out *Celsr2/3*

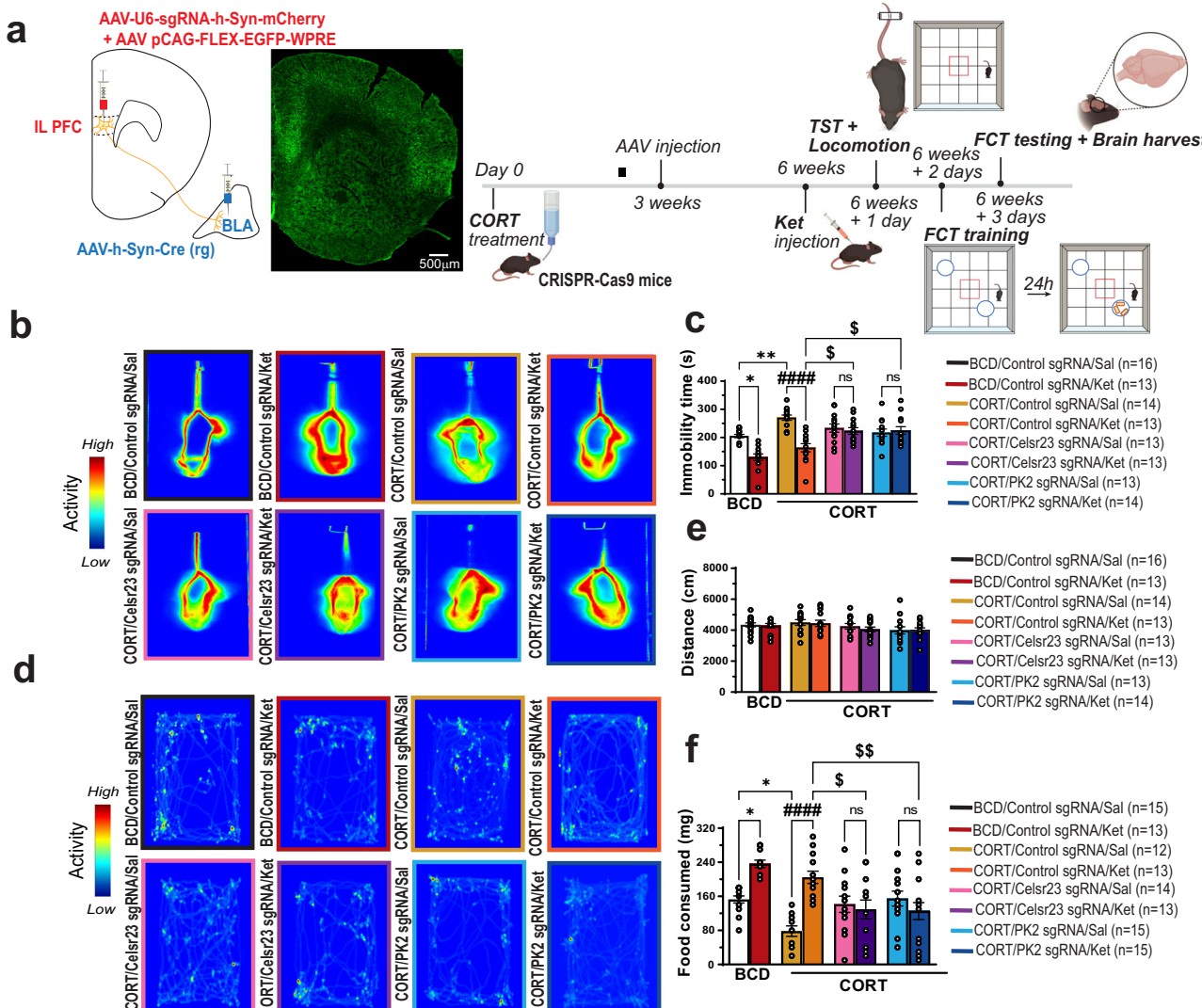

**Fig. 5 | *Celsr2/3* and *PK2* are required for the sustained antidepressant effect of Ket in chronically stressed mice in the BLA-projecting IL PFC neurons. a** Schematics of experimental design. The illustrations were created with BioRender. **b** Representative heatmaps of mouse movement in the TST. **c** Quantification of immobility time in the TST in (**a**) [$F_{(7, 101)} = 13.20$, $P < 0.0001$]. **d** Representative heatmaps of open field test. **e** Quantification of locomotion in the open field test in (**d**) [$F_{(7, 101)} = 1.321$, $P = 0.25$]. **f** Quantification of food consumption in the food

consumption test [$F_{(7, 102)} = 8.449$, $P < 0.0001$]. Each column represents the mean + S.E.M. of 12–16 animals. Statistical analysis was performed by one-way ANOVA followed by Tukey's test. *$p < 0.05$, **$p < 0.01$ compared with the control group (BCD/Control sgRNA) treated with saline, and ####$p < 0.0001$ compared with the CORT/Control sgRNA group treated with saline, $$p < 0.05$ and $$$$p < 0.01$ compared with the CORT/Control sgRNA group treated with ketamine.

and *PK2* prevented Ket from restoring the synapse numbers, indicating that these PCP proteins are required for Ket-induced synapse increase in this rodent model for chronic stress. In control animals not treated with CORT or ketamine, *Celsr2/3* and *PK2* conditional knockout led to a decrease of synapse numbers (Supplemental Fig. 7g–i), whereas conditional knockout of these PCP genes in control animals did not lead to behavioral changes (Supplemental Fig. 7a–f). Therefore, the lack of sustained antidepressant-like effect of ketamine is likely due to the loss of the ability of ketamine to induce new synapse formation in the absence of these PCP genes.

## Discussion

The discovery that a single subanesthetic dose of ketamine produces both acute and sustained antidepressant effects promises great potential for finding a cure[2]. Although the molecular and cellular targets and some of the mechanisms of the acute effects of ketamine have been better studied[21–23], the mechanisms underlying long-term antidepressant effects of ketamine (at least 7 days after the initial treatment) are much less known. The long-term antidepressant effects likely depend on long-term changes of circuits, including changes of synapse numbers. Our findings suggest that the planar cell polarity pathway mediates the ketamine-induced synapse restoration in IL PFC → BLA neurons, and provides a potential therapeutic strategy for synapse restoration without having to repeatedly use ketamine, which has undesirable side effects such as addiction and dissociation. Our results also highlight the distinct functions of different subtypes of mPFC neurons in mediating different aspects of depressive-like behaviors and the importance of synaptic restoration in behavioral remission by low-dose ketamine treatment.

The signaling pathway that directly regulates synapse assembly has been elusive. Recent findings suggest that PCP pathway is the key pathway for the formation and maintenance of the vast majority of glutamatergic synapses in the mammalian brain[7,9,10]. These previous studies established that the PCP components regulate synapse numbers in neural circuits that mediate cognitive functions. Here we show that the PCP components also regulate synapse numbers in neural

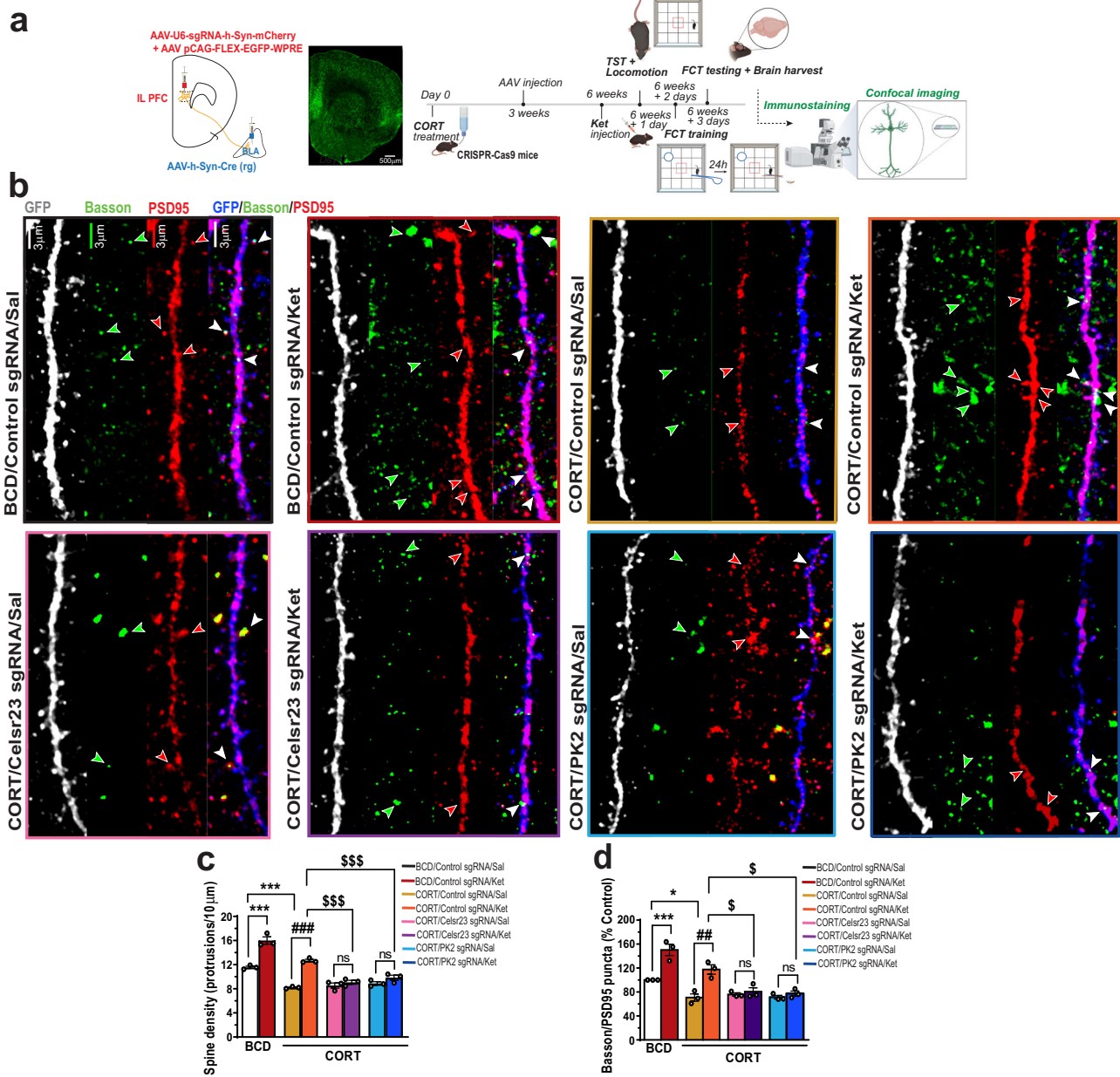

**Fig. 6 | Analyses of dendritic spine and excitatory synapses in the BLA-projecting IL PFC neurons. a** Schematics of experimental design. The illustrations were created with BioRender. **b** Representative micrographs showing dendritic spines and glutamatergic synapses in the BLA-projecting IL PFC neurons. **c** Quantification of dendritic spines [$F_{(7, 16)} = 56.68$, $P < 0.0001$]. **d** Quantification of glutamatergic synapses [$F_{(7, 16)} = 24.27$, $P < 0.0001$]. Each column represents the mean + S.E.M. of 3 animals (7-14 neurons/animal). Statistical analysis was performed by One-way ANOVA followed by Tukey's test. *$p < 0.05$, ***$p < 0.001$ compared with the control group (BCD/Control sgRNA) treated with saline, ##$p < 0.01$ and ###$p < 0.001$ compared with the CORT/Control sgRNA group treated with saline, $$p < 0.05$ and $$$$p < 0.001$ compared with the CORT/Control sgRNA group treated with ketamine. Scale bar 3 μm. Green arrowheads: Bassoon puncta. Red arrowheads: PSD95 puncta. White arrowheads: Bassoon/PSD95 puncta.

circuits that mediate emotional behaviors. By comparing our single-cell sequencing data with the human major depressive disorder (MDD) single nucleus RNAseq dataset, we found that that neuronal cell type-specific or neural circuit-specific changes of PCP genes are similar between this mouse model for chronic stress/depression and human MDD. It is worth testing whether PCP proteins may also mediate specific synaptic changes in human neurons or circuits using induced pluripotent stem cells or organoids derived from patients of MDD. In addition to the PCP proteins, our single-cell sequencing and cell–cell communication analyses also predicted changes of a number of other pathways upon corticosterone treatment and ketamine treatment, which may provide a starting point for future studies to furhter

understand the mechanisms underlying sustained anti-depressant effects of low-dose ketamine administration.

## Methods
### Animals
Male, 7 weeks old C57BL/6 mice (Stock #027; Charles River Labs), for the single-cell transcriptomics experiments, and male and female CRISPR/Cas9 knockin mice (Stock #026175; Jackson Laboratory), for all other experiments, were maintained at constant room with free access to water and food, under a 12:12 h light:dark cycle. The cages were placed in the experimental room for at least 1 h before the test for acclimatization. The procedures in this study were performed in

accordance with the NIH Guide for the Care and Use of Laboratory Animals and approved by the University of California, San Diego (UCSD) Institutional Animal Care and Use Committee. All efforts were made to minimize animal suffering and the number of animals used in the experiments.

## Drugs and treatments

**Chemogenetic inhibition.** C57BL/6 J mice were first anesthetized with isoflurane (vaporizer 2.5%) and then stereotactically injected bilaterally, 300 nl per injection site, into the IL PFC (#1: +1.5 mm anterior–posterior, ±0.35 mm medial-lateral, −3.13 mm dorsal–ventral; #2: +1.8 mm anterior–posterior, ±0.35 mm medial-lateral, −3.20 mm dorsal–ventral) with a Gi/o-coupled Cre-dependent inhibitory DREADDs hM4Di-mCherry virus (Stock #44362-AAV8; Addgene, Watertown, MA) or mCherry control (Stock #50459-AAV8; Addgene, Watertown, MA), and into the BLA (#1: −0.95 mm anterior–posterior, ±2.80 mm medial-lateral, −4.75 mm dorsal–ventral; #2: −1.32 mm anterior–posterior, ±2.80 mm medial-lateral, −4.50 mm dorsal–ventral; #3: −1.69 mm anterior–posterior, ±3.00 mm medial-lateral, −4.75 mm dorsal–ventral) or aIC (#1: +0.84 mm anterior–posterior, ±3.40 mm medial-lateral, −4.00 mm dorsal–ventral; #2: +1.34 mm anterior–posterior, ±2.80 mm medial-lateral, −3.75 mm dorsal–ventral; #3: +1.84 mm anterior–posterior, ±2.75 mm medial-lateral, −3.50 mm dorsal–ventral; #4: +2.10 mm anterior–posterior, ±2.50 mm medial-lateral, −3.50 mm dorsal–ventral) with AAV.hSyn.Cre.WPRE.hGH retrograde (Stock #105553-AAVrg; Addgene, Watertown, MA), 300 nl per injection site. Following three weeks post-surgery, mice were treated with the chemogenetic ligand compound 21 (C21, Stock #HB6124; Hello Bio, Princeton, NJ) at the dose of 2 mg/kg (i.p.) and 40 min later underwent a series of behavioral tests.

## Model of depression induced by chronic corticosterone exposure

Corticosterone, CORT (Stock #27840; Sigma, St. Louis, MO) was dissolved in 0.45% 2-hydroxypropyl cyclodextrin, BCD (Stock #332593; Sigma, St. Louis, MO) and delivered in lightproof bottles, available ad libitum in drinking water at a concentration of 35 µg/ml (equivalent to 5 mg/kg/day) for 6 weeks. CORT or vehicle control (0.45% BCD) was replaced every 2–3 days. Ketamine hydrochloride, Ket (Zetamine, Stock #501072; VetOne, Boise, ID) was dissolved in 0.9% sterile saline and administered via intraperitoneal route (i.p.) at a concentration of 10 mg/kg. The dissolution of the Ket was freshly done immediately before its administration. Appropriate vehicle treated groups (0.9% sterile saline) were also assessed simultaneously. All drugs were administered in a constant volume of 10 ml/kg body weight by an experienced investigator.

## Single-cell RNA-sequencing

For single-cell RNA-sequencing (scRNA-seq) experiments, C57BL/6 J mice were exposed to 35 µg/ml of CORT in the drinking water for 6 weeks. A control group received regular drinking water without CORT. After chronic CORT treatment, mice were treated with the fast-acting antidepressant Ket (10 mg/kg, i.p.). 24 h after Ket's treatment. Mice were then euthanized after being anesthetized in $CO_2$ chamber and brains were harvested, and the prefrontal cortices were rapidly dissected and transferred into ice-cold Hibernate A/B27 medium (60 ml Hibernate A medium with 1 ml B27 and 0.15 ml Glutamax). The tissues were enzymatically dissociated into single-cell suspension, and the cells were captured with the 10× Chromium platform (10× Genomics, Pleasanton, CA). Reverse transcription, cDNA amplification and library construction were performed according to the manufacturer's instructions and used for sequencing. The sequencing data were processed using the Cell ranger software to demultiplex the samples, generate FASTQ files, align the reads to mm9 genome and generate feature-barcode matrices. The filtered featured matrices were individually analyzed using Seurat R package (v3.1.1) and submitted to quality control (gene count per cell, percent of mitochondrial transcripts). Three to four data samples from the same experimental group (a total of 10 individual brain samples) were merged, normalized, clustered according to conserved cell type markers, and the differently expressed (DE) genes identified. For Gene Ontology (GO) analysis, each DE genes list was separated into upregulated and downregulated gene groups using log-fold change (logFC) as the delimitate. After the creation of these two group of genes (upregulated and downregulated genes), genes were uploaded into g:Profiler's functional profiling (g:GOSt) one experimental group at a time and the GO results generated.

## RNAscope multiplex fluorescent assay

For RNA detection within intact cells, an independent cohort of C57BL/6 J mice were exposed to 35 µg/ml of CORT in the drinking water for 6 weeks and treated with Ket (10 mg/kg, i.p.). 24 h after Ket's injection, brains were harvest, fresh-frozen in liquid nitrogen and cut into 10 µm coronal sections. The specimens were then fixed using 4% PFA and dehydrated through a graded series of increasing concentrations of ethanol. RNAscope multiplex fluorescent in situ hybridization was performed using RNAscope kit v2 (Advanced Cell Diagnostics, Newark, CA) following manufacturer's instructions. For fluorescent RNA imaging and quantification, the IL PFC was located, and z-stacks imaged at 40× magnification (Leica SP8 confocal). The number of transcripts/µm$^2$ was analyzed using ImageJ software (National Institute of Health, Bethesda, MD). Data regarding QC (negative and positive probes) are provided in Supplementary Fig. 5b and 5c, respectively.

## sgRNA design, expression analysis and virus generation

The Brie database was used for the design of sgRNAs. To validate the efficiency of sgRNAs candidates, individual sgRNA was cloned into PX549-SpCand as9 vector (Stock #62988; Addgene, Watertown, MA). Neuro2A cells were cultured on a 12-well plate. Cells were then transfected with 1 µg sgRNA plasmid by using 1 mg/ml Polyethyleneimine MAX (Polyscience). Puromycin was used to select the transfected cells. Genomic DNA of Neuro2A was purified by using the cloroform-ethonal method. The amplification of target DNA fragment and efficiency testing of individual sgRNAs were performed with manufacture's protocol of Surveyor assay.

Selected sgRNA sequences are as follows: Celsr2 sgRNA: 5′-GTA CACCGTTCGGCTCAACG-3′; Ceslr3 sgRNA: 5′-CGTTCGGGTGTTAT CAGCAC-3′; Prickle2 sgRNA: 5′-ACAGCCAGAGTCGTCATCTG-3′[9,10].

The sgRNAs were cloned into the AAV vector (Stock #87916; Addgene, Watertown, MA) for virus package. The LacZ sgRNA was used as a negative control. AAV was produced by transfection of AAV-293 cells with three plasmids: an AAV vector expressing target constructs, AAV helper plasmid (pHELPER; Agilent) and AAV rep-cap helper plasmid (pRC-DJ). Transfection of 293 cells was carried out using calcium phosphate. The cells were collected and lysed 72 h after transfection. Viral particles were purified by an iodixanol step-gradient ultracentrifugation method. qRT-PCR was used for virus titer measurement. The virus titer was ~1012 GC/mL[9,10].

## Labelling of neurons for analysis of dendritic spines and immunofluorescence staining

For labeling of dendritic spines and immunofluorescence staining of excitatory synapses, CRISPR–Cas9 mice were exposed to 35 µg/ml of CORT in the drinking water for three weeks. After the 3 weeks, mice were anesthetized with isoflurane (vaporizer 2.5%) and stereotactically injected bilaterally into the IL PFC (#1: +1.5 mm anterior–posterior, ±0.35 mm medial-lateral, −3.13 mm dorsal–ventral; #2: +1.8 mm anterior–posterior, ±0.35 mm medial-lateral, −3.20 mm dorsal–ventral) with a combination of AAV-FLEX-EGFP (Stock #51502; Addgene, Watertown, MA) and AAV-U6-sgRNA-h-Syn-mCherry, 300 nl per injection site; and into the BLA

(#1: −0.95 mm anterior−posterior, ±2.80 mm medial-lateral, −4.75 mm dorsal−ventral; #2: −1.32 mm anterior−posterior, ±2.80 mm medial-lateral, −4.50 mm dorsal−ventral; #3: −1.69 mm anterior−posterior, ±3.00 mm medial-lateral, −4.75 mm dorsal−ventral) with AAV.hSyn.-Cre.WPRE.hGH retrograde (Stock #105553-AAVrg; Addgene, Watertown, MA), 300 nl per injection site. Mice received analgesic treatment (Ethiqa extended release, 3.25 mg/kg, s.c.) to reduce pain and distress. Three weeks following surgery, mice were treated with Ket (10 mg/kg, i.p.) and 24 h later underwent a series of behavioral tests. Immediately after the behavioral tests, all animals were perfused with PBS followed by 4% PFA. Brains were removed and post-fixed in 4% PFA overnight at 4 °C. After, brains were cryoprotected in 30% sucrose for 2 days and coronal free-floating sections were prepared at 30 μm in a vibratome. The sections obtained were treated with 1% SDS for 5 min at room temperature for antigen retrieval, incubated in a blocking solution (1% bovine serum albumin, and 5% goat serum in Tris buffer saline solution (TBS) with 0.1% Triton X-100) for 1.5 h, and then stained overnight at 4 °C with primary antibodies chicken anti-GFP (Stock #ab13970; Abcam, Boston, MA), guinea pig anti-Bassoon (presynaptic marker, Stock #141004; Synaptic Systems, Goettingen, Germany), goat anti-PSD-95 (postsynaptic marker, Stock #ab12093; Abcam, Boston, MA), mouse anti-Iba-1 (microglia marker, Stock #MA5-27726; Invitrogen, Waltham, Massachusetts, USA), rabbit IL-1β (proinflammatory cytokine, Stock #PA5-105048; Invitrogen, Waltham, Massachusetts, USA), and rabbit anti-NF-κB (transcription factor that regulates inflammatory responses, Stock #51-0500; Invitrogen, Waltham, Massachusetts, USA). After, sections were incubated with fluorochrome-conjugated secondary antibodies Alexa 488 anti-chicken, Alexa 647 anti-guinea pig and Alexa 405 anti-goat solution for 2 h at room temperature and mounted in mounting media. Single-color images of the immunostaining in lower magnification provided in Supplementary Fig. 9. For dendritic spine density analysis, secondary and tertiary dendritic branches of layer 2/3 pyramidal neurons of the IL PFC cortex that project to the BLA were z-stacked and imaged at 63× magnification with 2× zoom-in (LSM510 Zeiss confocal microscope). Spine density analysis was performed using Bitplane Imaris software (Zurich, Switzerland). Protrusions not clearly defined or with lengths >5 μm were excluded from analysis. Three-dimensional view of dendritic spines provided in Supplementary Movie. S1. Excitatory synapse analysis was performed using ImageJ Synapse Counter plug-in (National Institute of Health, Bethesda, MD). When the pre- and post- are overlapped by 33–100%, they are considered colocalized. For inflammatory makers imaging and quantification, the IL PFC was located, and imaged at 20× magnification (Olympus VS200 Slide Scanner). The number of puncta was analyzed using ImageJ software (National Institute of Health, Bethesda, MD). All image acquisition and analyses were performed by experienced researchers blinded to the experimental conditions.

## Behavioral tests

**Tail suspension test (TST).** The total duration of immobility induced by tail suspension was measured using the method described by Steru et al.[17]. Acoustically and visually isolated mice were suspended 50 cm above the floor by adhesive tape placed approximately 1 cm from the tip of the tail (PanLab Harvard Apparatus, Holliston, MA, USA). Immobility time was recorded during a 6-min period by an experienced researcher[24–27]. TST was carried out in double blinded conditions and computally analyzed using SMART Video Tracking System (PanLab Harvard Apparatus, Holliston, MA, USA).

**Open-field test.** To assess the effects of treatments on locomotor activity, immediately after the tail suspension test, mice were evaluated in the open-field paradigm[24–27] by an experienced researcher double blinded to the experimental conditions. The apparatus was cleaned with a solution of 70% ethanol between tests to hide animal clues. The distance travelled by the mice was recorded in a 6-min session and computally analyzed using SMART Video Tracking System (PanLab Harvard Apparatus, Holliston, MA, USA).

**Food consumption test (FCT).** Moda-Sava et al.[8] found that 2 days after ketamine treatment, the deletion of newly formed spines blocked the long-term effects of ketamine on PFC activity events (calcium transients) and on motivated avoidance behavior in the TST but not in the SPT[8]. In our hand, Sucrose Preference Test did not show any difference after CORT treatment. (Supplementary Fig. 8a). To investigate feeding behavior (a symptom of depression which is defined as lack of feeling pleasure by common activities healthy subjects enjoy)[18], we designed a new test, which we called food consumption test (FCT). Twenty-four hour after the TST, animals were placed at the center of the open field arena in the presence of two identical petri dishes for a 5-min training session. Importantly, subjecting mice to a training session prior to the testing prevented any anxiety-related behavior that mice would experience by staying in a novel apparatus. Immediately after the training session, mice were deprived of food for 24 h and placed in the open field arena for the testing session in which regular food was added to one of the petri dishes. The food consumption was estimated by measuring the amount of food consumed during the 15 min testing session. Of note, mice food deprivation for FCT caused a reduction of less than 10% of mouse body weight (Supplemental Fig. 8b).

## Statistical analysis

Comparisons between multiple experimental groups were performed by one-way ANOVA followed by Tukey's HSD test, when appropriate. All statistical analyses were performed using GraphPad Prism software (La Jolla, California, USA). A value of $p < 0.05$ was considered significant.

## Reporting summary

Further information on research design is available in the Nature Portfolio Reporting Summary linked to this article.

## Data availability

The scRNA-seq data generated in this study have been deposited in the NIH GEO database under 580 accession code #GSE191016 (GEO Accession viewer (nih.gov)). All data presented in graphs 581 within the Figures in this study are provided in the Supplementary Information. Source data are provided as a Source Data file. Source data are provided with this paper.

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

## Acknowledgements
We thank Zou lab members for discussions and comments on the manuscript. In particular, we thank Yumna Azrar, Natasha Horne, Emma Kandel and Lauren Study for their practical help with mouse colony maintenance, mouse genotyping and brain sectioning. This work was supported by NIHM R01 MH116667 and NINDS R21 NS111648 grants to Y.Z. The images in Figs. 1a, 3a, b, 4a, 5a, 6a, and Supplementary Figs. 5a, 6a, and 7a were created with BioRender.

## Author contributions
Y.Z. and A.E.F. designed the study. A.E.F., S.G., B.F., C.B., and J.T. performed all experiments and analyzed the results. T.W. performed the transcriptomics and cell–cell signaling analyses. Y.B. provided AAV virus express sgRNA for Prickle2. A.B. provided insightful advice. Y.Z. and A.E.F. wrote the paper.

## Competing interests
Y.Z. is the founder of VersaPeutics and has equity, compensation and interim managerial role. The terms of this arrangement have been reviewed and approved by the University of California, San Diego in accordance with its conflict-of-interest policies. The remaining authors declare no competing interests.
