## [Peer Review File · Nature Communications]

Planar cell polarity proteins mediate ketamine-induced restoration of glutamatergic synapses in prefrontal cortical neurons in a mouse model for chronic stressREVIEWER COMMENTS

Reviewer #1 (Remarks to the Author):

In this study, Freitas et al use single cell transcriptomics to identify molecular pathways regulating synapse formation using the CORT-induced chronic stress model. Combined with morphology, behavior and calcium imaging analyses, the authors revealed that the PCP components underly synapse number changes mediating IL-BLA circuit-based depressive-like behavior as well as the long-term antidepressant effect of ketamine.

Given the role of synapse changes in behavior modification, the studies are informative for establishing a causal link. However, there are multiple problems concerning the methods, data presentation and interpretation that will have to be addressed before the study is publishable.

Major concerns:

A. Molecular study

1. Not enough details were provided for scRNA-seq experiment and analysis:

a) How many groups and how many mice in each group were used? What is the reproducibility of scRNA-seq? Any significant batch effect (cluster number, cell proportion) across different samples?

b) What are the criteria for DEG identification? What statistical test was used? What were the p-value and fold-change? The expression changes of *Celsr2/Celsr3/Prickle2* shown in Fig. 1e seem not significant.

c) What are the numbers of DEGs in each of the groups? It will be a good resource if a supplementary table containing the DEG information across different cell cluster can be added.

d) The GO analysis present in Fig. S2 lacks statistic evaluation, please add FDR information and move the results of the GO analysis of excitatory and inhibitory neurons to the main figure.

2. The authors stated to have sequenced 42,475 single cells from 10 independent samples. However, data presented in Fig.1b appears to have much fewer cells.

3. The DEGs related to regulating synapses and PCP genes exhibit cell type-specific changes in different treatment groups (Fig.S2, Fig.1). The authors should show some molecules known to be important in synapse formation to justify the importance of PCP study.

4. For the TST results (Fig. 3c), is there significant difference between BCD/Control sgRNA/Sal and the four knockout groups? If not, does it mean knocking out PCP genes could rescue the behavior phenotype caused by CORT treatment?

5. No evidence of Cas9-mediated knockout of target genes was provided. At least some level of verification, such as Surveyor assay or in vivo staining, should be provided.

B. Behavioral test

6. Sucrose preference test is the classic test for measuring anhedonia. The authors are suggested to try a lower concentration, such as 1% or 2% instead of 5%, to exclude the possibility of false-negative result. What's more, the Moda-Sava et al.' paper cannot serve as a strong rationale for switching to another test.

7. With regard to the food preference test, the authors trained the mice and deprived them from food for 24 hrs before testing their food consumptions. The authors must measure their body weight change (decrease ~10%) to make sure all the animals are still maintaining at a similar level of motivation because fasting can affect basic body homeostasis, which not only affect emotion and

executive behavior, but also motor function. Consequently, preference and food consumption cannot be used as a measurement for anhedonia without ruling out the other effects mentioned above. Applying high fat diet and regular chow without fasting to carry out the preference test for assessing anhedonia is highly recommended.

C. Fiber photometry

8. The authors should provide validation of viral expression (GCaMP) and position of cannula implantation.

9. More details of the fiber photometry study, including n numbers (3-8), should be provided. The authors should make it clear how many mice were used in each group for each experiment. Based on how variable the data is, 3 animals seem to be not enough. Besides the curves, direct comparison of different groups at specific time periods in the form of bar graphs with statistics is more informative.

10. The rationale of analyzing the Ca²⁺ signal of 1s before immobility and 8-12s after feeding onset is not clear. How could this be linked to behavioral phenotype? Is it possible to analyze the correlation of Ca²⁺ signal and behavior at the level of single animal? Because it is not clear how the 1s neuronal activity can be linked to the mouse behavior, a direct comparison of the neuronal activity before Ket and after Ket injection on the same animal might be informative. Plotting the signals of at least two "mobile-immobile" transition stages might be helpful to support their conclusion.

Minor points:

1. Switch Fig. 1d and Fig. 1e to match the order of description in the text.

2. Violin plot is not a good way to show differential expression between different groups (Fig. 1e), as in some case whether the difference is significant is not clear, which should be indicated in the figure. The author may only show data from excitatory and inhibitory neuron as those genes are barely expressed in other cell types.

3. The authors can consider splitting Fig. 1 into two containing scRNA-seq and RNAscope results, respectively.

4. Figure legend for Fig.3e is missing.

5. Lack of information on how the transcripts were quantified (pixel, threshold) in the RNAscope assay.

6. Annotation of x-axes should be aligned (Fig.1e)

Reviewer #2 (Remarks to the Author):

In this interesting and potentially high-impact work, Freitas and colleagues identify a role for planar cell polarity (PCP) genes in mediating the effects of ketamine on synapse formation and behavior in BLA-projecting infralimbic cortex neurons. This work has many strengths. The topic is timely and will appeal to this journal's general audience. The authors assemble data from a wide range of experimental sources to support their conclusions, including single cell transcriptomics, RNA scope, cell type-specific genetic knockouts, behavioral testing, and photometry. The experiments are well executed and carefully controlled. The manuscript is generally well written.

Overall, I am quite enthusiastic about this work and its potential for impact on the field. However, there are several issues I would recommend addressing before publishing it.

1. The focus on BLA-projecting IL cells is not unreasonable but is also not clearly justified. It would be helpful to evaluate key claims in another cell type that is hypothesized to be uninvolved in ketamine's antidepressant effects. Perhaps another brain region that is uninvolved in these behaviors or another projection subtype within IL. This would be valuable not only for evaluating the specificity of the IL => BLA projection and establishing that it plays a key role but also for controlling for other non-specific effects of the genetic knockdown. And as an endpoint, perhaps just the behavioral outcomes. (Certainly no need to redo every experiment in another cell type.)

2. The authors frame their work as testing "the causality between synapse number changes and behavioral changes." What the authors actually show is that targeted knockout of *Celsr23* or *PK2* is sufficient to block ketamine's effects on synapse number and on TST and food consumption behavior. They do not actually establish a direct causal link between synapse number and behavior because they are not manipulating synapse number directly. One way to address this would be to add control data (ephys? something else?) to evaluate the specificity of the genetic knockdown effect on synapse number and rule out non-specific effects on other physiological parameters. The experiment suggested in point 1 above would also be helpful. And lastly, it will probably be necessary to tone down claims about causal links between synapse formation and behavior, since they are not directly assessed here.

3. Transcriptomic data. In Figure 1 and Supp. Fig. 2, the authors analyze single cell sequencing data identify several biological pathways that are altered by CORT or ketamine. They then show that the expression of various PCP genes is altered in both the scRNA sequencing data and in an RNA scope experiment. It would be valuable to clarify how important these changes are in terms of effect size relative to other changes that are observed.

Relatedly and more generally, I felt the focus on PCP genes was well justified, but in general, the analysis of scRNA seq data seemed somewhat underdeveloped. How many genes are altered in each condition and to what extent? What is the relationship between genes altered after CORT treatment vs. after ketamine treatment? More could be done with the bioinformatics as well. As currently presented, the scRNAseq data feel tacked on and don't really add anything that isn't already demonstrated in the RNA scope experiment.

Lastly, transcriptomic data are typically uploaded to a repository and made publicly available at the time of publication. I didn't see any reference to plans for uploading their data to a repository, nor did I see a data availability statement. In the reporting checklist, the authors state that all data are available in the manuscript but that's not really standard anymore for transcriptomic datasets.

4. Figure presentation. While some of the figure panels were beautiful and very clearly presented, overall, I found many of them somewhat hard to follow. I think this had a lot to do with the very brief legends. Key details were sometimes hard to find. A few examples:

- Figure Supp. Fig. 2: The legend could be revised to more clearly explain the data. As is, it's short and leaves some questions unclear. Panel S2b is said to show pathways "downregulated by CORT" and S2c shows "pathways upregulated by ketamine." There is no description of how these pathways were identified statistically; whether this is a comprehensive description of **all** pathways meeting their criteria for being down- or unregulated, respectively or just a subset; whether there is overlap between what changed after CORT treatment and what changed after ketamine treatment; and whether there were also pathways regulated in the opposite direction, i.e. unregulated after CORT and or downregulated after ketamine.

- Fig. 1e: CORT and ketamine are described as respectively down- and up-regulating the expression of *Celsr2*, *Celsr3*, and *Prickle2* in the scRNA seq dataset but there are no statistics supporting this either in the figure or the legend or the main text. It's also hard to appreciate effect sizes.

- In 1f and 1g, it was not clearly described which bar graphs correspond to which cell types. Overall, it is quite unclear / hard to follow from the data as presented in Figure 1 which effects of CORT and ketamine are specific to which cell types. The main text makes claims about cell type specificity but this reviewer found it difficult to evaluate those claims based on the way the data were presented in Fig. 1. Wouldn't interaction statistics be needed to support claims of an effect in one cell type but not another?

5. Figure 3 behavior. It is true that the open field test is commonly used as a control for non-specific effects on locomotor activity as noted by the authors. However, the open field test is also used as a test of "anxiety-related behavior" by quantifying the amount of time a mouse spends in the center of the open field arena compared to the periphery. Did CORT, ketamine, and the sgRNA manipulations affect center time in the open field? This won't change the interpretation of the results but it would be valuable to include these data for comparison to other published works.

Also in Figure 3, it seems that the sgRNA manipulations not only altered the population mean for the immobility and food consumed measures but also affected the variance (they look much more highly variable than the other groups). Could this be due to mistargeting of the AAV or incomplete knockdown? It would be helpful to please include histology images validating the AAV targeting in Figs. 2 and 3 and visualizing the distribution of IL cells that project to BLA. It would also be helpful to include some validation data for the knockout (i.e. effects on Celsr23 and PK2 expression).

6. Photometry data. The data in Fig. 4 are interesting but somewhat hard to interpret and this is not really discussed. That is, the authors observe differences in activity in these cells as a function of experimental condition but the activity is not interpreted at all with respect to the behavior. What is the rationale for focusing on activity one second prior to immobility onset? What do the data look like with respect to the onset of struggling in the tail suspension test? It looks like activity increases during immobility in the BCD/saline mice and in the BCD/ketamine but not in the CORT/saline or CORT/ketamine groups. What does this mean? As is, the photometry experiments raise more questions than they answer.

MINOR:

- Please consider revising all figure legends to include more stats reporting, i.e. for each statistical test, please provide exact p values (not $P < X$), the value of the test statistic, df, etc, and where appropriate, perhaps some sense of effect size.

- Page 7: "The IL PFC-BLA neurons were shown to undergo changes of activity induced by stress and required for Ket-induced recovery." Up to this point in the paper, there have been no data speaking to changes in activity. Was this meant to refer to another publication?

- Intro: consider revising to clearly define the questions being asked in this study and any associated hypotheses

- Please consider avoiding anthropomorphic terms like "behavioral despair"

- Fig. 2a and Fig. 3a refer to the same experiment, right? It was slightly confusing that Fig. 2a referred to behavioral testing but there were no behavioral data until Figure 3.

- In Fig. 1b-c, inhibitory interneurons do not appear to be clearly differentiated from excitatory cells, at least not along the two dimensions plotted here. Some discussion might be useful.

Reviewer #3 (Remarks to the Author):

The paper by Dr Freitas and colleagues from the laboratory of Dr Zou examines the mechanism of ketamine reversal of a stress-related behavioral model in a mouse model. The authors conduct single cell analysis of gene expression changes after CORT and KET treatments. They identify changes in the expression of several planar cell polarity (PCP) genes that Dr Zou has previously shown to be involved in the regulation of excitatory synapses. The authors then test whether the effects of Ketamine depend on the Celsr proteins using inducible knockouts and CRISPR. The authors show that CORT-dependent effects on spine density fail to be reserved after removal of these proteins. In addition, selective targeting of PCP protein expression in the Pfc prevent rescue of CORT-dependent effects by ketamine. These findings are exciting and provide a link between PCP genes and stress induced models of depression. Less convincing are the authors arguments for the mechanism of action of these effects, eg changes in synapse formation number as causal. The paper would be significantly improved if the authors could demonstrate a more direct link between changes in spine density and the effects on calcium imaging or behavior that they observe.

In figure 2, the staining shown for the synaptic markers is not convincing. In most images show, there appear to be few if any puncta outside of the labeled dendrites, which would not be expected in brain sections, few spines have psd-95 or bassoon puncta, most of the synapses appear on the dendritic shaft rather than in the spines.

The data shown in figures 4 and 5 are noisy and not particularly convincing, particularly for the claims made by the authors that these experiments were conducted to examine the function of glutamatergic synapses. Using a bulk calcium signal which is the sum or all activity in the region lacks resolution raise questions about whether the authors can assign the effects they observe to a particular type of synapse. Additional experiment would be needed to show that the changes observed are due to changes in synapse number directly rather than changes in neuronal excitability, calcium entry, or a presynaptic effect. These experiments could be conducted using optogenetic tools targeted to the pathways the authors suggest are involved. Key tests would be to show reduced optically evoked activation in the PFC after CORT that is rescued by ket and blocked by knocking out PCP components. Moreover, missing controls are demonstration of the selective labeling of the PFC, targeting of the fiber and fraction/type of cells labeled.

Minor

In figures 3-5, the authors indicate the behavior using a very light yellowish box, this is very difficult to see. The onset and end of the behavior should be more clearly indicated. A color not found in the traces should be used.

The paper appears to be formatted for a different journal. The authors should consider increasing the details provided in the introduction and discussion.

REVIEWER COMMENTS

Reviewer #1 (Remarks to the Author):

In this study, Freitas et al use single cell transcriptomics to identify molecular pathways regulating synapse formation using the CORT-induced chronic stress model. Combined with morphology, behavior and calcium imaging analyses, the authors revealed that the PCP components underly synapse number changes mediating IL-BLA circuit-based depressive-like behavior as well as the long-term antidepressant effect of ketamine.

Given the role of synapse changes in behavior modification, the studies are informative for establishing a causal link.

We thank the reviewer for the positive comment.

However, there are multiple problems concerning the methods, data presentation and interpretation that will have to be addressed before the study is publishable.

We address the concerns below.

Major concerns:

A. Molecular study

*1. Not enough details were provided for scRNA-seq experiment and analysis:
a) How many groups and how many mice in each group were used? What is the reproducibility of scRNA-seq? Any significant batch effect (cluster number, cell proportion) across different samples?*

We have now included the details about the scRNA-seq experiments and analyses in the revised manuscript.

b) What are the criteria for DEG identification? What statistical test was used? What were the p-value and fold-change? The expression changes of Celsr2/Celsr3/Prickle2 shown in Fig. 1e seem not significant.

In the revised manuscript, we are using CellChat to analyze change of cell-cell communication (Fig. 2). And the changes we show are significant. We moved the DEG results, which show a trend of change but are not significant, to supplementary Fig. 4

c) What are the numbers of DEGs in each of the groups? It will be a good resource if a supplementary table containing the DEG information across different cell cluster can be added.

We moved the DEG results, which show a trend of change but are not significant, to supplementary Fig. 4

d) The GO analysis present in Fig. S2 lacks statistic evaluation, please add FDR information and move the results of the GO analysis of excitatory and inhibitory neurons to the main figure.

In the revised manuscript, we do not include GO analyses. Instead, we used CellChat analyses, which are described in the methods section.

2. The authors stated to have sequenced 42,475 single cells from 10 independent samples. However, data presented in Fig.1b appears to have much fewer cells.

We performed quality control and the data used for analyses are from 13,381 high quality cells.

3. The DEGs related to regulating synapses and PCP genes exhibit cell type-specific changes in different treatment groups (Fig.S2, Fig.1). The authors should show some molecules known to be important in synapse formation to justify the importance of PCP study.

In the revised manuscript, we are using CellChat to analyze changes of cell-cell communication and found that the PCP pathway showed significant changes (Fig. 2).

4. For the TST results (Fig. 3c), is there significant difference between BCD/Control sgRNA/Sal and the four knockout groups? If not, does it mean knocking out PCP genes could rescue the behavior phenotype caused by CORT treatment?

We thank the reviewer for this question. We performed these experiments and found that PCP gene (Celsrs and PK2) knockout did not lead to changes of behavior during the time frame of our experiments (Supplemental Fig. 6a-6f).

5. No evidence of Cas9-mediated knockout of target genes was provided. At least some level of verification, such as Surveyor assay or in vivo staining, should be provided.

We have previous verified and reported the Cas9-mediated knockout in our earlier publications. To confirm the cell type specific knocked indeed worked, we performed staining and reported the results in Supplemental Fig. 6.

B. Behavioral test

6. Sucrose preference test is the classic test for measuring anhedonia. The authors are suggested to try a lower concentration, such as 1% or 2% instead of 5%, to exclude the possibility of false-negative result. What's more, the Moda-Sava et al.' paper cannot serve as a strong rationale for switching to another test.

We agree with reviewer's opinion that SRT is a classic test for measuring anhedonia. However, like described by Moda-Sava et al, we did not observe any effect on SRT after corticosterone treatment either (Supplemental Figure 8a). Therefore, we developed the "food preference test", which was able to observe a difference after corticosterone treatment. We decided to call that food consumption test rather than food preference test as we are not comparing two kinds of food.

7. With regard to the food preference test, the authors trained the mice and deprived them from food for 24 hrs before testing their food consumptions. The authors must measure their body weight change (decrease ~10%) to make sure all the animals are still maintaining at a similar level of motivation because fasting can affect basic body homeostasis, which not only affect emotion and executive behavior, but also motor function. Consequently, preference and food consumption cannot be used as a measurement for anhedonia without ruling out the other effects mentioned above. Applying high fat diet and regular chow without fasting to carry out the preference test for assessing anhedonia is highly recommended.

We appreciate the suggestions of the reviewer and measured their body weight change and indeed the reduction was around 7.89% (Supplemental Figure 8b). We have also decided to name it food consumption test rather than "food preference test", as we are not actually making any comparisons.

C. Fiber photometry

8. The authors should provide validation of viral expression (GCaMP) and position of cannula implantation.

Due to the lack of clarify of how the Ca⁺ signaling from fiber photometry correlates to behavior and whether Ca⁺ signaling is a direct reflection of synapse numbers, we decided that the fiber photometry experiment after knocking out PCP genes is not very informative and relevant. We decided to not to include the fiber photometry experiment.

9. More details of the fiber photometry study, including n numbers (3-8), should be provided. The authors should make it clear how many mice were used in each group for each experiment. Based on how variable the data is, 3 animals seem to be not enough. Besides the curves, direct comparison of different groups at specific time periods in the form of bar graphs with statistics is more informative.

Due to the lack of clarify of how the Ca⁺ signaling from fiber photometry correlates to behavior and whether Ca⁺ signaling is a direct reflection of synapse numbers, we decided that the fiber photometry experiment after knocking out PCP genes is not very informative and relevant. We decided to not to include the fiber photometry experiment.

10. *The rationale of analyzing the Ca²⁺ signal of 1s before immobility and 8-12s after feeding onset is not clear. How could this be linked to behavioral phenotype? Is it possible to analyze the correlation of Ca²⁺ signal and behavior at the level of single animal? Because it is not clear how the 1s neuronal activity can be linked to the mouse behavior, a direct comparison of the neuronal activity before Ket and after Ket injection on the same animal might be informative. Plotting the signals of at least two “mobile-immobile” transition stages might be helpful to support their conclusion.*

Due to the lack of clarify of how the Ca⁺ signaling from fiber photometry correlates to behavior and whether Ca⁺ signaling is a direct reflection of synapse numbers, we decided that the fiber photometry experiment after knocking out PCP genes is not very informative and relevant. We decided to not to include the fiber photometry experiment.

Minor points:

1. *Switch Fig. 1d and Fig. 1e to match the order of description in the text.*

The figures have been updated.

2. *Violin plot is not a good way to show differential expression between different groups (Fig. 1e), as in some case whether the difference is significant is not clear, which should be indicated in the figure. The author may only show data from excitatory and inhibitory neuron as those genes are barely expressed in other cell types.*

We moved the violin plot to Supplemental Figure 4. Gene expression changes are shown by RNAscope in the main figure.

3. *The authors can consider splitting Fig. 1 into two containing scRNA-seq and RNAscope results, respectively.*

The figures have been updated.

4. *Figure legend for Fig.3e is missing.*

The figures have been updated.

5. *Lack of information on how the transcripts were quantified (pixel, threshold) in the RNAscope assay.*

The number of pixels were quantified and normalized to saline control.

6. *Annotation of x-axis should be aligned (Fig. 1e)*

The figures have been updated.

Reviewer #2 (Remarks to the Author):

In this interesting and potentially high-impact work, Freitas and colleagues identify a role for planar cell polarity (PCP) genes in mediating the effects of ketamine on synapse formation and behavior in BLA-projecting infralimbic cortex neurons. This work has many strengths. The topic is timely and will appeal to this journal's general audience. The authors assemble data from a wide range of experimental sources to support their conclusions, including single cell transcriptomics, RNA scope, cell type-specific genetic knockouts, behavioral testing, and photometry. The experiments are well executed and carefully controlled. The manuscript is generally well written.

We appreciate the highly positive comments from the reviewer.

Overall, I am quite enthusiastic about this work and its potential for impact on the field. However, there are several issues I would recommend addressing before publishing it.

1. The focus on BLA-projecting IL cells is not unreasonable but is also not clearly justified. It would be helpful to evaluate key claims in another cell type that is hypothesized to be uninvolved in ketamine's antidepressant effects. Perhaps another brain region that is uninvolved in these behaviors or another projection subtype within IL. This would be valuable not only for evaluating the specificity of the IL => BLA projection and establishing that it plays a key role but also for controlling for other non-specific effects of the genetic knockdown. And as an endpoint, perhaps just the behavioral outcomes. (Certainly no need to redo every experiment in another cell type.)

We thank the reviewer for the suggestion. In Figure 3, we compared the insular cortex projecting IL neurons with the BLA projecting neurons and showed that the IL=> BLA neurons are specifically involved in depressive-like behavior.

2. The authors frame their work as testing "the causality between synapse number changes and behavioral changes." What the authors actually show is that targeted knockout of Celsr23 or PK2 is sufficient to block ketamine's effects on synapse number and on TST and food consumption behavior. They do not actually establish a direct causal link between synapse number and behavior because they are not manipulating synapse number directly. One way to address this would be to add control data (ephys? something else?) to evaluate the specificity of the genetic knockdown effect on synapse number and rule out non-specific effects on other physiological parameters. The experiment suggested in point 1 above would also be helpful. And lastly, it will probably be necessary to tone down claims about causal links between synapse formation and behavior, since they are not directly assessed here.

We appreciate the criticism. We therefore rewrote the entire paper and no longer frame our work as testing "the causality between synapse number changes and behavioral changes". We do provide the quantification of synapse numbers in Figure 6 as we are showing targeted knockout of Celsr23 or PK2 in the BLA

projecting infra-limbic PFC neurons is sufficient to block ketamine's effects on synapse number and on TST and food consumption behavior. Related to this, we decided not to include the fiber photometry experiment because Ca^{2+} signaling is a direct reflection of synapse numbers.

3. *Transcriptomic data.* In Figure 1 and Supp. Fig. 2, the authors analyze single cell sequencing data identify several biological pathways that are altered by CORT or ketamine. They then show that the expression of various PCP genes is altered in both the scRNA sequencing data and in an RNA scope experiment. It would be valuable to clarify how important these changes are in terms of effect size relative to other changes that are observed.

In the revised manuscript, we used CellChat to analyze changes of cell-cell signaling instead of gene ontology pathway analyses and gene expression changes. CellChat is a much more powerful method.

Relatedly and more generally, I felt the focus on PCP genes was well justified, but in general, the analysis of scRNA seq data seemed somewhat underdeveloped. How many genes are altered in each condition and to what extent? What is the relationship between genes altered after CORT treatment vs. after ketamine treatment? More could be done with the bioinformatics as well. As currently presented, the scRNAseq data feel tacked on and don't really add anything that isn't already demonstrated in the RNA scope experiment.

We performed more complete analyses and used CellChat to analyze changes of cell-cell signaling. CellChat is a much more powerful method.

Lastly, transcriptomic data are typically uploaded to a repository and made publicly available at the time of publication. I didn't see any reference to plans for uploading their data to a repository, nor did I see a data availability statement. In the reporting checklist, the authors state that all data are available in the manuscript but that's not really standard anymore for transcriptomic datasets.

The scRNAseq data were uploaded to NIH repository and will be released from the repository upon publication.

4. *Figure presentation.* While some of the figure panels were beautiful and very clearly presented, overall, I found many of them somewhat hard to follow. I think this had a lot to do with the very brief legends. Key details were sometimes hard to find. A few examples:

- *Figure Supp. Fig. 2: The legend could be revised to more clearly explain the data. As is, it's short and leaves some questions unclear. Panel S2b is said to show pathways "downregulated by CORT" and S2c shows "pathways upregulated by ketamine." There is no description of how these pathways were identified statistically; whether this is a comprehensive description of *all* pathways meeting their criteria for being down- or*

unregulated, respectively or just a subset; whether there is overlap between what changed after CORT treatment and what changed after ketamine treatment; and whether there were also pathways regulated in the opposite direction, i.e. unregulated after CORT and or downregulated after ketamine.

Those figures are no longer included as we have updated figures.

*- Fig. 1e: CORT and ketamine are described as respectively down- and up-regulating the expression of *Celsr2*, *Celsr3*, and *Prickle2* in the scRNA seq dataset but there are no statistics supporting this either in the figure or the legend or the main text. It's also hard to appreciate effect sizes.*

We moved the violin plot to Supplemental Figure 4. Gene expression changes are shown by RNAscope in the main figure.

- In 1f and 1g, it was not clearly described which bar graphs correspond to which cell types. Overall, it is quite unclear / hard to follow from the data as presented in Figure 1 which effects of CORT and ketamine are specific to which cell types. The main text makes claims about cell type specificity but this reviewer found it difficult to evaluate those claims based on the way the data were presented in Fig. 1. Wouldn't interaction statistics be needed to support claims of an effect in one cell type but not another?

We replaced those figures with CellChat analyses.

5. Figure 3 behavior. It is true that the open field test is commonly used as a control for non-specific effects on locomotor activity as noted by the authors. However, the open field test is also used as a test of "anxiety-related behavior" by quantifying the amount of time a mouse spends in the center of the open field arena compared to the periphery. Did CORT, ketamine, and the sgRNA manipulations affect center time in the open field? This won't change the interpretation of the results but it would be valuable to include these data for comparison to other published works.

We did not observe any changes in center/surround time.

*Also in Figure 3, it seems that the sgRNA manipulations not only altered the population mean for the immobility and food consumed measures but also affected the variance (they look much more highly variable than the other groups). Could this be due to mistargeting of the AAV or incomplete knockdown? It would be helpful to please include histology images validating the AAV targeting in Figs. 2 and 3 and visualizing the distribution of IL cells that project to BLA. It would also be helpful to include some validation data for the knockout (i.e. effects on *Celsr23* and *PK2* expression).*

We have now included histology images validating AAV targeting and visualizing the IL=>BLA neurons. We performed the validation of KO and included the results in Supplemental Figure 6.

6. *Photometry data. The data in Fig. 4 are interesting but somewhat hard to interpret and this is not really discussed. That is, the authors observe differences in activity in these cells as a function of experimental condition but the activity is not interpreted at all with respect to the behavior. What is the rationale for focusing on activity one second prior to immobility onset? What do the data look like with respect to the onset of struggling in the tail suspension test? It looks like activity increases during immobility in the BCD/saline mice and in the BCD/ketamine but not in the CORT/saline or CORT/ketamine groups. What does this mean? As is, the photometry experiments raise more questions than they answer.*

Due to the lack of clarify of how the Ca⁺ signaling from fiber photometry correlates to behavior and whether Ca⁺ signaling is a direct reflection of synapse numbers, we decided that the fiber photometry experiment after knocking out PCP genes is not very informative and relevant. We decided to not to include the fiber photometry experiment.

MINOR:

- Please consider revising all figure legends to include more stats reporting, i.e. for each statistical test, please provide exact p values (not $P < X$), the value of the test statistic, df, etc, and where appropriate, perhaps some sense of effect size.

Statistic data have been provided.

- Page 7: "The IL PFC-BLA neurons were shown to undergo changes of activity induced by stress and required for Ket-induced recovery." Up to this point in the paper, there have been no data speaking to changes in activity. Was this meant to refer to another publication?

Due to the lack of clarify of how the Ca⁺ signaling from fiber photometry correlates to behavior and whether Ca⁺ signaling is a direct reflection of synapse numbers, we decided that the fiber photometry experiment after knocking out PCP genes is not very informative and relevant. We decided to not to include the fiber photometry experiment.

- Intro: consider revising to clearly define the questions being asked in this study and any associated hypotheses

We revised.

- Please consider avoiding anthropomorphic terms like "behavioral despair"

We revised.

- Fig. 2a and Fig. 3a refer to the same experiment, right? It was slightly confusing that

Fig. 2a referred to behavioral testing but there were no behavioral data until Figure 3.

We revised to clarify.

- In Fig. 1b-c, inhibitory interneurons do not appear to be clearly differentiated from excitatory cells, at least not along the two dimensions plotted here. Some discussion might be useful.

We revised the analyses.

Reviewer #3 (Remarks to the Author):

The paper by Dr Freitas and colleagues from the laboratory of Dr Zou examines the mechanism of ketamine reversal of a stress-related behavioral model in a mouse model. The authors conduct single cell analysis of gene expression changes after CORT and KET treatments. They identify changes in the expression of several planar cell polarity (PCP) genes that Dr Zou has previously shown to be involved in the regulation of excitatory synapses. The authors then test whether the effects of Ketamine depend on the Celsr proteins using inducible knockouts and CRISPR. The authors show that CORT-dependent effects on spine density fail to be reserved after removal of these proteins. In addition, selective targeting of PCP protein expression in the Pfc prevent rescue of CORT-dependent effects by ketamine. These findings are exciting and provide a link between PCP genes and stress induced models of depression. Less convincing are the authors arguments for the mechanism of action of these effects, eg changes in synapse formation number as causal. The paper would be significantly improved if the authors could demonstrate a more direct link between changes in spine density and the effects on calcium imaging or behavior that they observe.

We appreciate the highly positive comments from the reviewer.

In figure 2, the staining shown for the synaptic markers is not convincing. In most images show, there appear to be few if any puncta outside of the labeled dendrites, which would not be expected in brain sections, few spines have psd-95 or bassoon puncta, most of the synapses appear on the dendritic shaft rather than in the spines.

We have updated the figure to include better images with synaptic markers (Figure 7).

The data shown in figures 4 and 5 are noisy and not particularly convincing, particularly for the claims made by the authors that these experiments were conducted to examine the function of glutamatergic synapses. Using a bulk calcium signal which is the sum or all activity in the region lacks resolution raise questions about whether the authors can assign the effects they observe to a particular type of synapse. Additional experiment would be needed to show that the changes observed are due to changes in synapse number directly rather than changes in neuronal excitability, calcium entry, or a presynaptic effect. These experiments could be conducted using optogenetic tools

targeted to the pathways the authors suggest are involved. Key tests would be to show reduced optically evoked activation in the PFC after CORT that is rescued by ket and blocked by knocking out PCP components. Moreover, missing controls are demonstration of the selective labeling of the PFC, targeting of the fiber and fraction/type of cells labeled.

Due to the lack of clarify of how the Ca⁺ signaling from fiber photometry correlates to behavior and whether Ca⁺ signaling is a direct reflection of synapse numbers, we decided that the fiber photometry experiment after knocking out PCP genes is not very informative and relevant. We decided to not to include the fiber photometry experiment.

Minor

In figures 3-5, the authors indicate the behavior using a very light yellowish box, this is very difficult to see. The onset and end of the behavior should be more clearly indicated. A color not found in the traces should be used.

We have revised the Figures.

The paper appears to be formatted for a different journal. The authors should consider increasing the details provided in the introduction and discussion.

We revised and provided more details.

REVIEWER COMMENTS

Reviewer #1 (Remarks to the Author):

In the revision, the authors have put significant efforts in addressing some of my questions, including comparing cell-cell communication changes after ketamine treatment and they also added data on PFC-aIC circuit.

Despite these improvements, the impact of the study was significantly weakened due to the removal of DEGs analysis and fiber photometry recording, which I asked for in the first round of review. Rather than addressing my questions, the authors choose to remove the sections, which raised my concern on whether the progress made in the current manuscript is enough to justify its publication in Nature Communication. In addition, the current manuscript still have a number of issues in experiment design and data analysis detailed below:

1. The PFC-BLA projection in feeding behavior modulation has been previously reported (PMID: 24441680). In this study, the authors chose to use food consumption to measure anhedonia, which is neither new nor appropriate because food consumption is mainly driven by energy homeostasis. Previously the authors claimed that there is no difference in sucrose preference, if true, the authors should be careful when mention anhedonia.
2. This study has only sequenced 13,381 high-quality cells from 10 independent samples. The total cell numbers are relatively low for identifying neuronal subtypes. Indeed, only 3 Inhibitory and 2 Excitatory subtypes were identified, which failed to reflect the neuronal heterogeneity in mPFC. In addition, the percentage of neuronal cells (~ 20% in fig.1b) is much lower than previous publications (> 50%), which raises concern on the quality of their scRNA-seq library.
3. The authors omitted DEGs analysis, an important criteria for identifying transcriptional alterations. While I commend the authors for incorporating cell-cell communication analysis, which examines expression changes in ligand-receptor pairs, it should be noted that this doesn't serve as a complete replacement for DEGs analysis. Additionally, it's difficult to discern the cell type information from Fig. 1d and Fig. 2 for cell-cell communication.
4. In Fig1c, what the authors listed are genes. How these genes were identified? If they emphasize the pathways the genes involved, please present the analysis. What are the criteria in ranking "top" pathways? What is the rationale in analyzing IL-1 β and NF κ B and how these two genes were chosen as markers? It seems the two markers are not related to their further study.
5. In Fig.3a right panel, the site of virus expression is not in the IL region, which raises concern whether they are truly targeting the IL-aIC circuit. Injection sites need to be presented. Additionally, the authors should replace Fig.5a and Fig.6a with higher quality images.
6. The authors need to pay more attention in writing. For the description of Fig.3, they want to change Food preference test (FPT) to food consumption test (FCT). However, they did not change the diagram of Fig.3b nor writing in the result. The same issue exists in Fig.5 and 6 as well.
7. Fig.1 e-h were not mentioned in results.

Reviewer #2 (Remarks to the Author):

The authors have done an outstanding job of responding to all of my comments. I particularly appreciate the addition of data from insula-projecting IL neurons and the new "CellChat" analyses. This is an exceptionally interesting series of findings and I have no further questions.

Reviewer #3 (Remarks to the Author):

The authors have made substantial revisions to their manuscript, addressing many of the initial concerns raised by the reviewers. These revisions include changing the data analysis approach for their sequencing data and moving away from their calcium imaging experiments. However, it is unclear why they have not included the Gene Ontology (GO) Analysis, at least as supplemental information. Are these data inconsistent with the Cell-Chat data? Given the concerns about the consistency of the GO data, it would be important to provide more information regarding Cell-Chat validation or demonstrate that two different data analysis approaches yield similar information. This could be included as supplemental data.

Another significant change was the removal of data from their calcium imaging experiments, rather than attempting to improve these experiments. To compensate for this loss, they have provided more analysis of their expression data. Given the interest in the mechanism of action of ketamine, this change is likely acceptable, but it does somewhat lessen the excitement for this project.

There are a couple of additional concerns raised by the changes made to the manuscript:

In Figure 6, the authors present data from staining for synaptic markers. However, the staining for these markers is uneven outside of the labeled neurons. The authors should provide single-color images of the immunostaining for these experiments. Additionally, lower magnification images should be included in the supplemental data.

The authors claim that overlap between Bassoon and PSD-95 is shown as white pixels. However, these are not apparent in the images presented in Figure 6. The authors need to provide clearer examples. This issue could also be addressed by showing the individual images for Bassoon and PSD-95 separately.

Given that knockouts of the studied genes reduce synapse density, the experiments with ketamine are somewhat difficult to interpret. Are the effects due to a steady-state decrease in synapse density or a lack of response to the drug? At a minimum, this should be discussed as a potential limitation of the study.

Minor:

For the immunostaining data, the authors should add a key to the figure to help readers understand what is being shown with each color

REVIEWER COMMENTS

Reviewer #1 (Remarks to the Author):

In the revision, the authors have put significant efforts in addressing some of my questions, including comparing cell-cell communication changes after ketamine treatment and they also added data on PFC-aIC circuit.

We thank the reviewer for the positive feedback.

Despite these improvements, the impact of the study was significantly weakened due to the removal of DEGs analysis and fiber photometry recording, which I asked for in the first round of review. Rather than addressing my questions, the authors choose to remove the sections, which raised my concern on whether the progress made in the current manuscript is enough to justify its publication in Nature Communication. In addition, the current manuscript still have a number of issues in experiment design and data analysis detailed below:

We would like to clarify that the CellChat, a recently available quantitative tool to analyze the expression of ligand-receptor pairs, is based on the DEG analysis and particularly suitable for assessing changes of cell-cell communication. During our revision, we tested CellChat and found that the changes of the PCP pathway was statistically significant. Therefore, we felt that CellChat, which quantifies both the ligands and the receptors, is a better way to assess changes than the DEGs of individual PCP components. As we stated in our Response to Reviewers in the first revision, the differences that DEGs only in our single cell sequencing data showed a trend but did not reach statistical significance, likely due to the lower number of single cells in our dataset. Therefore, we moved the DEG data to the supplementary figure. Now with CellChat, we were also able to analyze the Major Depressive Disorder (MDD) Single Nucleus RNAseq Human Dataset downloaded from the UCSC cell browser, which included a total of over 160,000 nuclei from 71 dorsolateral-PFC samples. Celsr2 and Celsr3 were downregulated in most types of excitatory neurons in humans (Fig 2b). Like in mouse mFPC, Celsr2 was increased between excitatory neurons and inhibitory interneurons (PV, ADARB2). Interestingly, Celsr2 was found decreased between excitatory neurons and VIP interneurons in the human dataset. These analyses suggest that neuron cell type-specific or neural circuit specific changes of PCP proteins are strikingly similar between the mouse model for chronic stress/depression and human MDD and may mediate specific synaptic changes in these neurons or circuits in humans.

In the previous round of revision, Reviewers 2 and 3 correctly pointed out that the Ca⁺⁺ signals measures by fiber photometry do not directly reflect synapse numbers as Ca⁺⁺ reflects overall neural activity including synaptic strength. Therefore, we agree that the fiber photometry experiment after knocking out PCP genes is not directly relevant. We decided to not to include the fiber photometry

experiment. Instead, we performed new chemogenetic data and found that the mPFC→ BLA circuit, but not the mPFC→ aIC circuit, mediates depressive-like behaviors and that synapse restoration mediated by the PCP proteins in the mPFC→ BLA circuit is required for the sustained anti-depressive effect of low dose ketamine. We think these new data adds to the knowledge about the distinct functions of different subtypes of mPFC neurons and demonstrate the importance and mechanism of synaptic restoration in behavioral remission by low-dose ketamine. We added additional comments in the Discussion section to highlight the significance of these findings.

1. The PFC-BLA projection in feeding behavior modulation has been previously reported (PMID: 24441680). In this study, the authors chose to use food consumption to measure anhedonia, which is neither new nor appropriate because food consumption is mainly driven by energy homeostasis. Previously the authors claimed that there is no difference in sucrose preference, if true, the authors should be careful when mention anhedonia.

We thank the reviewer for this comment. We have rephrased the part where we mention the food consumption results.

2. This study has only sequenced 13,381 high-quality cells from 10 independent samples. The total cell numbers are relatively low for identifying neuronal subtypes. Indeed, only 3 Inhibitory and 2 Excitatory subtypes were identified, which failed to reflect the neuronal heterogeneity in mPFC. In addition, the percentage of neuronal cells (~20% in fig.1b) is much lower than previous publications (> 50%), which raises concern on the quality of their scRNA-seq library.

We agree with the reviewer about the limitations of our dataset, which is not large. In order to ensure quality, we eliminated the cells that did not meet the quality. Our goal was to use gene expression approach to screen for candidate molecular pathways to study further. We were able to discern interesting pathways that may be relevant, including the PCP pathway. Indeed, neuron cell type-specific or neural circuit specific changes of PCP proteins we found in mouse are strikingly similar to human MDD, which somewhat validates our dataset.

3. The authors omitted DEGs analysis, an important criteria for identifying transcriptional alterations. While I commend the authors for incorporating cell-cell communication analysis, which examines expression changes in ligand-receptor pairs, it should be noted that this doesn't serve as a complete replacement for DEGs analysis. Additionally, it's difficult to discern the cell type information from Fig. 1d and Fig. 2 for cell-cell communication.

We would like to clarify that the CellChat, a recently available quantitative tool to analyze the expression of ligand-receptor pairs, is based on the DEG analysis and particularly suitable for assessing changes of cell-cell communication. During our revision, we tested CellChat and found that the changes of the PCP pathway

was statistically significant. Therefore, we felt that CellChat, which quantifies both the ligands and the receptors, is a better way to assess changes than the DEGs of individual PCP components. As we stated in our Response to Reviewers in the first revision, the differences that DEGs only in our single cell sequencing data showed a trend but did not reach statistical significance, likely due to the lower number of single cells in our dataset. Therefore, we moved the DEG data to the supplementary figure.

We increased the font size in Fig. 1d and Fig. 2.

4. In Fig1c, what the authors listed are genes. How these genes were identified? If they emphasize the pathways the genes involved, please present the analysis. What are the criteria in ranking “top” pathways? What is the rationale in analyzing IL-1 β and NF κ B and how these two genes were chosen as markers? It seems the two markers are not related to their further study.

CDH and VTN are the names of the pathways by CellChat that were reduced most by corticosterone but increased the most following ketamine administration of the corticosterone treated animals in mPFC. SEMA6, EGF, NRG, L1CAM, NEGR, NRXN, VCAM, CADM, CNTN and NCAM are the pathways that were reduced most by corticosterone but increased the most following ketamine administration of the corticosterone treated animals in BLA. We found that many of the same cell-cell communication pathways that were downregulated in corticosterone treated mice were upregulated after ketamine treatment. Most of the pathways which showed the greatest reversal in the corticosterone-treated animals by ketamine treatment are cell-cell interacting molecules, including cell adhesion pathways that may impact neuronal morphology and structure. Therefore, we decided to highlight these pathways in Fig.1c.

IL-1 β and NF κ B staining in Figure 1e-1h was meant to further test whether our experimental system to induce stress with corticosterone and reduce stress with ketamine has worked. If so, we can further evaluate the expression of PCP genes in Figure 2 using our dataset. It is known that stress and depression induce inflammatory responses and our CellChat analyses also suggest that (Fig. 1d). We further investigated our gene expression data by gene ontology using genes that had an absolute average Log₂ fold change greater than 0.1 and found that more than 50 different significantly changed signaling pathways underwent changes of gene expression (top 10 pathways are listed in Supplemental Fig. 2b, 2c). NF κ B signaling in the MAPK pathway was found to have increased in corticosterone (ranked top 5) and then decreased after ketamine treatment as shown in the Kegg diagrams (ranked top 10) (Supplemental Fig. 2D, 2E). Therefore, we analyzed the levels of IL-1 β and NF κ B in microglia in Figure 1e-1h.

5. In Fig.3a right panel, the site of virus expression is not in the IL region, which raises concern whether they are truly targeting the IL-aIC circuit. Injection sites need to be presented. Additionally, the authors should replace Fig.5a and Fig.6a with higher quality images.

We thank the reviewer for his/her comment about the IL-aIC circuit. Please note that Fig. 3a right panel has been updated. We are also showing the aIC injection site below:

Finally, Fig.5a and Fig.6a have been replaced with higher quality images.

6. The authors need to pay more attention in writing. For the description of Fig.3, they want to change Food preference test (FPT) to food consumption test (FCT). However, they did not change the diagram of Fig.3b nor writing in the result. The same issue exists in Fig.5 and 6 as well.

We thank the reviewer for raising this point. The figures and manuscript have been updated accordingly.

7. Fig.1 e-h were not mentioned in results.

We have updated the results sections to mention Fig.1 e-h results.

Reviewer #2 (Remarks to the Author):

The authors have done an outstanding job of responding to all of my comments. I particularly appreciate the addition of data from insula-projecting IL neurons and the new "CellChat" analyses. This is an exceptionally interesting series of findings and I have no further questions.

We appreciate the positive feedback.

Reviewer #3 (Remarks to the Author):

The authors have made substantial revisions to their manuscript, addressing many of the initial concerns raised by the reviewers. These revisions include changing the data analysis approach for their sequencing data and moving away from their calcium imaging experiments. However, it is unclear why they have not included the Gene Ontology (GO) Analysis, at least as supplemental information. Are these data inconsistent with the Cell-Chat data? Given the concerns about the consistency of the GO data, it would be important to provide more information regarding Cell-Chat validation or demonstrate that two different data analysis approaches yield similar information. This could be included as supplemental data.

In the updated version of our manuscript, we have opted for Cell Chat analysis over GO analysis. This decision was based on CellChat's greater relevance for our objective of comparing the PCP pathway, as it is a cell-cell communication pathway and is the main focus of our paper. According to our findings, CellChat's predictions align well with our GO analysis.

Another significant change was the removal of data from their calcium imaging experiments, rather than attempting to improve these experiments. To compensate for this loss, they have provided more analysis of their expression data. Given the interest in the mechanism of action of ketamine, this change is likely acceptable, but it does somewhat lessen the excitement for this project.

In the previous round of revision, Reviewers 2 and 3 correctly pointed out that the Ca⁺⁺ signals measures by fiber photometry does not directly reflect synapse numbers as Ca⁺⁺ reflects overall neural activity including synaptic strength. Therefore, we agree that the fiber photometry experiment after knocking out PCP genes is not directly relevant. We decided to not to include the fiber photometry experiment. Instead, we performed new chemogenetic data and found that the mPFC→ BLA circuit, but not the mPFC→ aIC circuit, mediates depressive-like behaviors and that synapse restoration mediated by the PCP proteins in the mPFC→ BLA circuit is required for the sustained anti-depressive effect of low dose ketamine. We think these new data adds to the knowledge about the distinct functions of different subtypes of mPFC neurons and demonstrate the importance and mechanism of synaptic restoration in behavioral remission by low-dose ketamine. We added additional comments in the Discussion section to highlight the significance of these findings

As this reviewer noted, we did provide more analyses of our gene expression data, particularly with CellChat for cell-cell communication. This new analyses are not only more relevant to our main focus on planar cell polarity proteins, which is a cell-cell communication pathway but also allowed us to compare with human MDD data. We also think that CellChat analyzes is more suitable for our paper than general GO analyses, which are essentially similar analyses but about pathways beyond cell-cell communication.

There are a couple of additional concerns raised by the changes made to the manuscript:

In Figure 6, the authors present data from staining for synaptic markers. However, the staining for these markers is uneven outside of the labeled neurons. The authors should provide single-color images of the immunostaining for these experiments. Additionally, lower magnification images should be included in the supplemental data.

We thank the reviewer for the comment. Single-color images of the immunostaining have now been provided, accordingly; as well as, lower magnification images in the supplemental data.

The authors claim that overlap between Bassoon and PSD-95 is shown as white pixels. However, these are not apparent in the images presented in Figure 6. The authors need to provide clearer examples. This issue could also be addressed by showing the individual images for Bassoon and PSD-95 separately.

We appreciate the feedback. Clearer examples of Bassoon and PSD-95 have been provided as individual single-color images.

Given that knockouts of the studied genes reduce synapse density, the experiments with ketamine are somewhat difficult to interpret. Are the effects due to a steady-state decrease in synapse density or a lack of response to the drug? At a minimum, this should be discussed as a potential limitation of the study.

Based on our results, we propose that PCP proteins may be needed for making new synapses induced by ketamine administration, which are required for behavior remission. When we knocked out the PCP genes in animals that were not treated with corticosterone or ketamine, we did not observe changes of behavior but we observed a reduction of synapse numbers. It is possible that when we knocked out the PCP genes, we were reducing all synapses, whether they were active (functional) or not, or even selectively those that were not active. Therefore, behaviors were not affected. Therefore, we think the lack of ketamine effects in PCP gene knockout is due to the lack of response in the absence of PCP genes as the knockout itself does not cause depressive-like behavior. We now include this discussion in the text.

Minor:

For the immunostaining data, the authors should add a key to the figure to help readers understand what is being shown with each color

We have updated the figures by adding a key indicating what is being shown with each color.

REVIEWERS' COMMENTS

Reviewer #1 (Remarks to the Author):

The authors have provided some broadly satisfactory response to my questions. I understand some of the modifications that they have made is to make the manuscript fit the scope of the study. Especially using CellChat and moving the DEGs to supplementary. However, I would recommend that the authors to include some of their observations in discussion:

1. Quantifications obtained from CellChat, although significant, are still 'predictive'. Future biological experiments will eventually show whether this indeed is true.
2. Corticosterone provides a model for stress-induced depression in mouse. While it provides a tractable model for depression, and correlates with transcriptional changes in human MDD, future studies are needed to determine whether this is indeed the mechanism in MDD.

Reviewer #3 (Remarks to the Author):

The authors have addressed the formatting issues with their figures, added some discussion to the manuscript that addresses some of my other concerns, and indicated that they would not address other issues. Overall, this is an interesting manuscript that adds to our understanding of planar cell polarity proteins in a model of stress.

Reviewer #1 (Remarks to the Author):

The authors have provided some broadly satisfactory response to my questions. I understand some of the modifications that they have made is to make the manuscript fit the scope of the study. Especially using CellChat and moving the DEGs to supplementary. However, I would recommend that the authors to include some of their observations in discussion:

1. Quantifications obtained from CellChat, although significant, are still 'predictive'. Future biological experiments will eventually show whether this indeed is true.

We agree it is important to remind the readers that CellChat is a predictive tool. We added that point in Discussion and also indicated the CellChat analyses were “predicted”, “predictive” or “inferred” throughout the text.

2. Corticosterone provides a model for stress-induced depression in mouse. While it provides a tractable model for depression, and correlates with transcriptional changes in human MDD, future studies are needed to determine whether this is indeed the mechanism in MDD.

We added in the discussion that future studies will be needed to test the PCP proteins in human MDD patients.

Reviewer #3 (Remarks to the Author):

The authors have addressed the formatting issues with their figures, added some discussion to the manuscript that addresses some of my other concerns, and indicated that they would not address other issues. Overall, this is an interesting manuscript that adds to our understanding of planar cell polarity proteins in a model of stress.

We thank the reviewer for the positive comments and the constructive suggestions throughout the review and revision process.